# Inverse centrifugal effect induced by collective motion of vortices in rotating thermal convection

Shan-Shan Ding [1,5], Kai Leong Chong[2,4,5], Jun-Qiang Shi[1], Guang-Yu Ding [2,3], Hao-Yuan Lu[1], Ke-Qing Xia [2,3 ✉] & Jin-Qiang Zhong [1 ✉]

When a fluid system is subject to strong rotation, centrifugal fluid motion is expected, i.e., denser (lighter) fluid moves outward (inward) from (toward) the axis of rotation. Here we demonstrate, both experimentally and numerically, the existence of an unexpected outward motion of warm and lighter vortices in rotating thermal convection. This anomalous vortex motion occurs under rapid rotations when the centrifugal buoyancy is sufficiently strong to induce a symmetry-breaking in the vorticity field, i.e., the vorticity of the cold anticyclones overrides that of the warm cyclones. We show that through hydrodynamic interactions the densely distributed vortices can self-aggregate into coherent clusters and exhibit collective motion in this flow regime. Interestingly, the correlation of the vortex velocity fluctuations within a cluster is scale-free, with the correlation length being proportional ($\approx 30\%$) to the cluster length. Such long-range correlation leads to the counterintuitive collective outward motion of warm vortices. Our study brings insights into the vortex dynamics that are widely present in nature.

[1] School of Physics Science and Engineering, Tongji University, Shanghai, 200092, China. [2] Department of Physics, The Chinese University of Hong Kong, Shatin, Hong Kong, China. [3] Center for Complex Flows and Soft Matter Research and Department of Mechanics and Aerospace Engineering, Southern University of Science and Technology, Shenzhen 518055, China. [4] Present address: School of Mechanics and Engineering Science, Shanghai University, Shanghai 200072 PR, China. [5] These authors contributed equally: Shan-Shan Ding, Kai Leong Chong. ✉email: xiakq@sustech.edu.cn; jinqiang@tongji.edu.cn

Coherent vortex structures exist ubiquitously in many flow systems ranging from small-scale turbulence to large-scale geophysical and astrophysical flows[1–3], and their dynamics play a crucial role in determining turbulent mixing and transport in those systems. Previous studies of vortex dynamics are mainly focused on isolated vortices[3,4]. However, in rapidly rotating turbulent flows these vortices become densely distributed[5,6], and the resulting vortex interactions may lead to markedly different dynamics compared to that of isolated vortices[7–9]. Many nonequilibrium dynamical systems in nature consisting of densely distributed, interacting entities often exhibit collective behavior, i.e., the entities self-aggregate to perform collective motions. Examples include bird flocks, bacteria swarms, and clustering of active matters[10–12]. Whether the collective behavior of vortices can arise in rotating turbulent flows is thus a question of fundamental interest.

The fluid dynamics of rotating turbulent flows is often studied in rotating Rayleigh–Bénard convection (RBC). Despite the considerable progress achieved in studying non-rotating and weakly rotating turbulent RBC[13–18], some important convection regimes that may exhibit intriguing vortex dynamics are yet to be explored (see, e.g.,[19,20]). Recent studies report that for rapidly rotating RBC in moderate Prandtl-number fluids, the convective flows are organized by the Coriolis force into coherent columnar vortices[21–30]. These columnar vortices are helical structures with either upward or downward flows. Upwelling vortices rotate in the same direction as the system (cyclones) in the lower half fluid layer, and in the opposite direction (anticyclones) in the upper half and vice versa for downwelling ones[22,29]. The theory of thermal wind balance, which relates the vertical variations of the fluid velocities with the horizontal temperature gradients[31], provides a general description of the flow structures of the columnar vortices. From a dynamical viewpoint, however, the horizontal motions of, and the interactions between, these vortices remain to be explored in a quantitative way.

Here, we demonstrate both experimentally and numerically the collective motion of vortices in rotating thermal convection. As a primary external force governing the motions of rotating fluids in many natural and industrial flows[19,32,33], centrifugal force drives cold, denser fluid radially outward from the rotation axis and warm, lighter fluid inward. Counterintuitively, we discover that the long-range correlated vortex dynamics give rise to inverse centrifugal motion, i.e., the warm and lighter convective vortices exhibit outward motion from the rotation axis. This intriguing phenomenon occurs in a rapidly rotating regime where the strong centrifugal buoyancy breaks the symmetry in both the population and vorticity magnitude of the vortices. Our study reveals that it is through local hydrodynamic interactions that the densely distributed vortices self-aggregate into large-scale vortex clusters, in which the warm cyclonic vortices submit to the collective motion dominated by the strong anticyclones and move outwardly.

## Results

**Vortex motion and inverse centrifugation.** Our experimental apparatus was designed for high-precision flow structure measurements in rotating RBC[20,30]. We used cylindrical cells with an inner diameter $d = 240$ mm and height $H = 63.0$ (120.0) mm, yielding an aspect ratio $\Gamma = d/H = 3.8$ (2.0). The experiment was conducted with a constant Prandtl number $\mathrm{Pr} = \nu/\kappa = 4.38$ and in the range $2.0 \times 10^7 \leq \mathrm{Ra} \leq 2.7 \times 10^8$ of the Rayleigh number $\mathrm{Ra} = \alpha g \Delta T H^3/(\kappa \nu)$ ($g$ is the gravitational acceleration, $\Delta T$ the applied temperature difference, $\alpha$, $\kappa$ and $\nu$ are respectively the isobaric thermal expansion coefficient, thermal diffusivity, and kinematic viscosity of the convecting fluid). Rotation rates of up to 5.0 rad/s were used. Thus the Ekman number $\mathrm{Ek} = \nu/(2\Omega H^2)$ spanned $4.9 \times 10^{-6} \leq \mathrm{Ek} \leq 2.7 \times 10^{-4}$, corresponding to a range of the reduced Rayleigh number $1.3 \leq \mathrm{Ra}/\mathrm{Ra}_c \leq 166$. Here, $\mathrm{Ra}_c = C\mathrm{Ek}^{-4/3}$ (with the coefficient $C = 8.7 - 9.63\mathrm{Ek}^{1/6}$) is the critical value for the onset of convection[34], and $\Omega$ is the rotating rate. The Froude number $\mathrm{Fr} = \Omega^2 d/(2g)$ varied within $0 < \mathrm{Fr} \leq 0.31$. The flow field at a fluid depth of $z = H/4$ was measured using the technique of particle image velocimetry (PIV) (see a schematic of the experimental setup in Fig. 1i). In the direct numerical simulation (DNS) we solved the Navier–Stokes equations with the Coriolis and centrifugal forces included, using the multiple-resolution version of the CUPS code[35,36]. The simulation was performed in a cylindrical domain with $\Gamma = 4$,

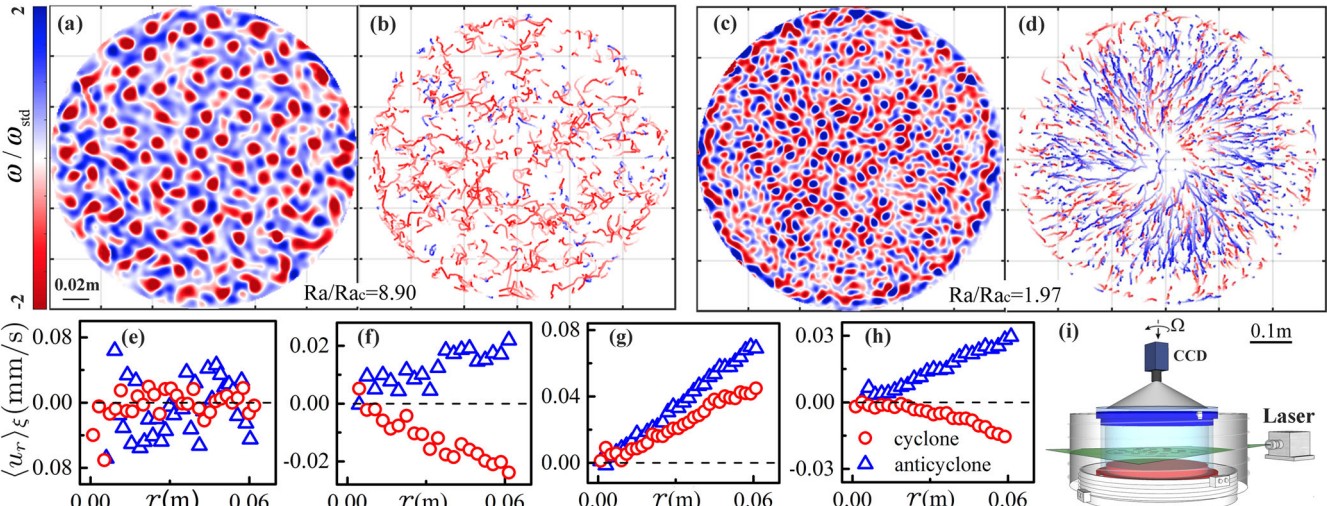

**Fig. 1 Experimental apparatus and radial motion of vortices. a, c** Instantaneous vertical vorticity distribution $\omega/\omega_{std}$ over the measured fluid height. $\omega_{std}$ is the standard deviation of $\omega$. **b, d** Trajectories for cyclones (red) and anticyclones (blue). The shading of the trajectories indicates that the vortices appear (terminate) at the light (dark)-color side. Results for $\mathrm{Ra} = 3.0 \times 10^7$ and $\mathrm{Ra}/\mathrm{Ra}_c = 8.90$, $\mathrm{Fr} = 0.03$ (**a, b**); $\mathrm{Ra}/\mathrm{Ra}_c = 1.97$, $\mathrm{Fr} = 0.27$ (**c, d**). **e–h** Radial profiles $\langle u_r(r) \rangle_\xi$ of cyclones and anticyclones. Data for $\mathrm{Ra} = 2.0 \times 10^7$ and from left to right, $\mathrm{Ra}/\mathrm{Ra}_c = 20.9, 4.57, 2.26, 1.43$, corresponding to the four flow regimes (I)–(IV), respectively (see text for discussions). Here, $\xi$ denotes individual vortex trajectory and $\langle...\rangle_\xi$ a trajectory-ensemble average. (**i**) Schematic of the experimental set-up. A laser sheet illuminates a rotating Rayleigh–Bénard convection cell filled with water and seeded with tracer particles at a fluid height $z = H/4$. A co-rotating camera images the light scattered by the tracer particles.

$Ra = 2.0 \times 10^7$, and $1.3 \leq Ra/Ra_c \leq 55$ (see Supplementary Fig. 1 for a phase diagram of the present study).

Figure 1 presents snapshots of the vortex structures at the measurement fluid height ($z = H/4$). At a low rotation rate when the centrifugal force is negligible (Fig. 1a), the cyclonic vortices (shown in red color) possess a greater number density and on average larger vorticity in magnitude than that of anticyclones (blue color). Both types of vortices exhibit stochastic horizontal motions as evidenced by the vortex trajectories shown in Fig. 1b. The mean-square-displacement (MSD) of the vortices becomes a linear function of time at large times, indicating a Brownian-type, normal diffusive motion[20].

However, at higher rotation rates when the centrifugal force becomes dominant, we observe strong anticyclones with a larger population than the cyclones (Fig. 1c). The anticyclones undergo outward radial motions accompanied by stochastic fluctuations along their paths (Fig. 1d) until they move close to the sidewall where their radial motion is terminated by the retrogradely traveling plumes. Compared to the anticyclones, the motion of weak cyclones is much more complex. Figure 1d indicates that in the outer region ($r \geq d/4$), the cyclones move toward the cell center while in the inner region ($r \leq d/4$) most of them migrate radially outward. In this rapidly rotating case, the MSD of both types of vortices indicates superdiffusive behavior (see Supplementary Movies of the vortex motions).

To further quantify the vortex motions, we show the profiles of the mean radial velocity $\langle u_r \rangle_\xi$ of the vortices measured in the inner region of the cell in Fig. 1e–h. These velocity profiles reveal four distinct flow regimes depending on the rotation rates: (I) A randomly diffusive regime exists in the slow rotating limit with Ra being one order in magnitude larger than $Ra_c$. In this regime the vortices move in a random manner, yielding $\langle u_r \rangle_\xi \approx 0$ (Fig. 1e). (II) A centrifugation-influenced regime where the magnitude of $\langle u_r \rangle_\xi$ increases linearly with $r$ ($4Ra_c \leq Ra \leq 10Ra_c$). We observe that warm cyclones (cold anticyclones) move radially inward (outward), which is in agreement with the centrifugal effect (Fig. 1f). (III) Inverse-centrifugal regime ($1.6Ra_c \leq Ra \leq 4 Ra_c$) in which there is anomalous outward cyclonic motion (Fig. 1g), and the radial gradients of $\langle u_r \rangle_\xi$ for both types of vortices reach a maximum. (IV) The asymptotic regime in the rapid rotation limit ($Ra \leq 1.6Ra_c$) where the opposite radial motions of cyclones and anticyclones recover (Fig. 1h).

We formulate a theoretical model consisting of Langevin-type equations that incorporate the centrifugal force, which governs the radial vortex motion in a background of stochastic fluctuations. As shown in Fig. 2, the model provides predictions of the first and second moments of the radial vortex displacements which replicate very well the experimental data in flow regimes (II) and (IV) (see Supplementary Note 3 for detailed discussions of the model). Nonetheless, Fig. 2 clearly shows that the inverse centrifugal motion of the cyclones in the anomalous regime (III) cannot be explained by the model. A key question is then what sets the anomalous vortex motion?

**Asymmetric vorticity field in the inverse-centrifugal regime**. To gain insights into the observed phenomenon, we first examine the relative strength of vorticity between the cyclones and anticyclones. Figure 3a shows the vorticity ratio $\gamma_\omega$ of the anticyclones to the cyclones. We note that in the randomly diffusive regime, $\gamma_\omega$ is approximately 0.6, meaning that at the measurement height $z = H/4$ the cyclonic vorticity is overall larger in magnitude than the anticyclonic ones (see Fig. 1a). It is the case because when observed at the lower half of the layer, anticyclones are downwelling vortices generated from the top boundary. They travel a long distance to the measurement layer than the upwelling vortices (cyclones), and their momentum and vorticity have been largely dissipated by the background turbulence when reaching the measurement position in this flow regime[27]. (The vorticity magnitude of the two types of vortices are equal if measured at $z = H/2$. See the visualization from our DNS in the right inset of Fig. 3a.) With increasing $\Omega$ the up- and down-welling vortices evolve into columnar structures that are vertically antisymmetric in vorticity with respect to the mid-height plane[25,26]. One would expect that in this flow regime the measured vorticity strength of the cyclones and anticyclones become comparable, i.e., $\gamma_\omega$ approaches unity. Our DNS data with the centrifugal force switched off indeed show this trend. However, when the centrifugal effect is dominant, both the experimental and DNS results reveal that $\gamma_\omega$ exceed unity considerably in the inverse-centrifugal regime, indicating an asymmetric vorticity field dominated by the cold anticyclones (left inset of Fig. 3a). In the asymptotic regime where the severe rotational constraint finally weakens the convective vortices, $\gamma_\omega$ eventually returns to unity, and the symmetry of the cyclonic and anticyclonic vorticity restores. Remarkably, we observe that $\gamma_\omega(Ra/Ra_c)$ is independent of Ra and $\Gamma$ over the parameter range studied.

We now show that the asymmetry of the vorticity field in the anomalous regime results from the centrifugal effect. Figure 3b presents the radial profiles of the mean temperature of the two kinds of vortices and of the background fluid. These numerical data indicate noticeable warming of the background fluid in the inner region, owing to the centrifugal effect[37–39]. (See comparative results in Supplementary Fig. 7 when the centrifugal force is excluded.) As a result, the temperature difference $\delta T$ of the cold anticyclones from the background exceeds that of the warm cyclones. Since $\delta T$ is proportional to the buoyancy forcing on the vortices, it is predicted to be positively correlated to the vorticity

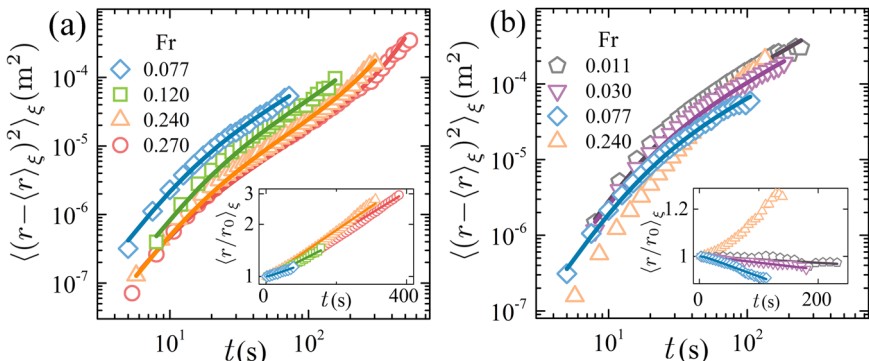

**Fig. 2 Second moments of vortex radial displacement.** Results for anticyclones (**a**) and cyclones (**b**) with $Ra = 3.0 \times 10^7$. Open symbols: experimental data. Solid lines: theoretical predictions. Insets: results of the first moments of vortex radial displacement. $r_0$ is the initial radial position of a vortex.

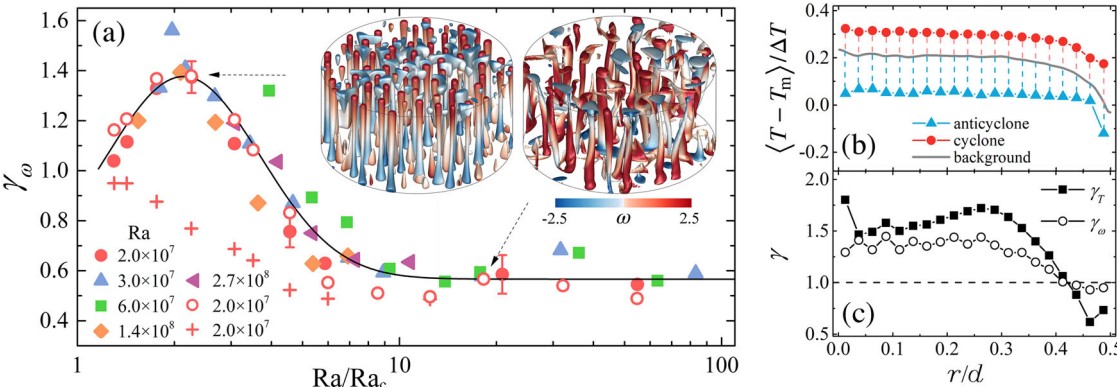

**Fig. 3 Vorticity and temperature of cyclones and anticyclones. a** The vorticity ratio $\gamma_\omega = |\langle\omega_a\rangle/\langle\omega_c\rangle|$ of the anticyclones to the cyclones as a function of Ra/Ra$_c$. Here, $\langle...\rangle$ denotes a time average. Filled symbols: experimental data for $\Gamma = 3.8$ with Ra $= 2.0 \times 10^7$ (circles), $3.0 \times 10^7$ (up triangles), $6.0 \times 10^7$ (squares); and for $\Gamma = 2.0$ with Ra $= 1.4 \times 10^8$ (diamonds), $2.7 \times 10^8$ (left triangles). Data from DNS including (excluding) the centrifugal force are shown in open diamonds (pluses) for $\Gamma = 4.0$. The solid curve indicates the trend of the experimental data. The error bars denote representative fluctuation amplitude of $\gamma_\omega$. Inset panels: iso-surfaces of the temperature anomaly from DNS for Ra $= 2.0 \times 10^7$, Ra/Ra$_c = 2.26$ (left), and Ra/Ra$_c = 18.3$ (right). The coloration represents the vorticity of the vortices. **b** Radial profiles of the mean temperature $\langle T - T_m\rangle/\Delta T$ for the vortices and the background fluid. $T_m$ is the mean of the top and bottom fluid temperature. The length of the dashed lines indicates the temperature difference $\delta T$ between the cyclones (anticyclones) and the background fluid. **c** Radial profiles of $\gamma_\omega$ and $\gamma_T = |\langle\delta T_a\rangle/\langle\delta T_c\rangle|$. **b**, **c** DNS data for Ra $= 2.0 \times 10^7$ and Ra/Ra$_c = 2.26$.

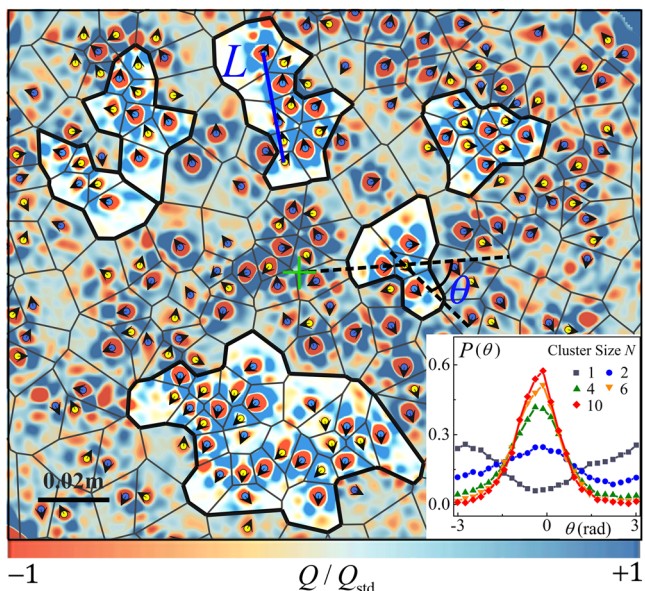

**Fig. 4 Clustering of vortices.** Results are shown in the central region for Ra $= 3.0 \times 10^7$, Ra/Ra$_c = 1.97$, Fr $= 0.27$. Blue (yellow) circles show the centers of anticyclones (cyclones). Black arrows show the vortex velocity direction. The solid-line network represents the Voronoi diagram of the vortex centers. Examples of six vortex clusters are highlighted and marked with thick boundaries. $\theta$ denotes the angle between the position vector of a cyclone relative to the rotation axis (green cross) and its velocity. $L$ is the largest distance between two vortices within a cluster. The background coloration represents the distribution of the quantity $Q/Q_{std}$. Inset: Probability density functions of $\theta$ of cyclones in clusters with various size $N$.

$\omega$ in recent theoretical models[24,26]. Indeed Fig. 3c shows that $\delta T$ and $\omega$ are both larger in magnitudes for anticyclones than for cyclones in the inner region, which explains the asymmetry of the vorticity field ($\gamma_\omega > 1$).

**Clustering and collective motion of the vortices.** In light of the broken symmetry of the vorticity field in the anomalous regime, we show below that it is the long-range correlated vortex motion that gives rise to the inverse centrifugal motion of the cyclones.

Figure 4 presents the instantaneous motion of the vortices, with their spatial distribution presented in a Voronoi diagram. One sees that the adjacent vortices often self-organize into vortex clusters, i.e., the vortices move largely in the same direction. We adopt two criteria to identify vortex clusters, i.e., the distance of two neighboring vortices is smaller than 1.5 times the mean vortex diameter and the angle $\phi$ between their velocity vectors is within a threshold ($\phi \leq \phi^* = 60°$). Our analysis over the range $30° \leq \phi^* \leq 75°$ confirms that the results of correlated vortex motion are not sensitive to the choice of $\phi^*$. The direction of the motion of each cyclone $i$ is represented by the angle $\theta$ between its velocity $\vec{u}_i$ and its position vector $\vec{r}_i$ from the rotation axis. We find that $\theta$ is strongly dependent on the number ($N$) of vortices in a cluster (inset). For isolated cyclones ($N = 1$), the most probable direction of motion is radially inward ($\theta_p = \pi$). However, for clustered cyclones ($N > 1$) we find $\theta_p = 0$ as they move outward. The standard deviation of $\theta$ decreases monotonically when $N$ increases. Our data reveal that within large clusters the motion of weak cyclones submit to that of strong anticyclones and move outwardly in a collective manner. Their inverse centrifugal motion becomes more unidirectional with the increase of the cluster size.

By analyzing the cluster size distribution $p(N)$, one can obtain certain insights into the physical mechanism responsible for the vortex cluster formation. Figure 5a shows that $p(N)$ for various Ra/Ra$_c$ can be well described by $p(N) = AN^{-b}e^{-N/N_c}$. Here, $b$ and $N_c$ are the fitting parameters with their dependence of Ra/Ra$_c$ plotted in the inset. For clusters in small size $N$, $p(N)$ first decays as a power function $N^{-b}$ up to a cutoff size $N_c$. Studies of the collective behavior in various natural systems have revealed that local aggregation of interacting entities is the essential ingredient for the power-law decay of the group-size distributions in these systems[40–42]. In the present vortex system, each vortex is surrounded more likely by counter-rotating vortices in a densely populated state (Fig. 4). Owing to the vortex-pair interaction, adjacent vortices of opposite-sign tend to move in similar directions[43] (see Supplementary Note 7 for details). Moreover, the shielded structure formed near the edge of each vortex prevents strong interactions in closer proximity[26,28,30], thus avoiding vortex merging and annihilation. As a result, isolated vortices are often aggregated into neighboring clusters and move

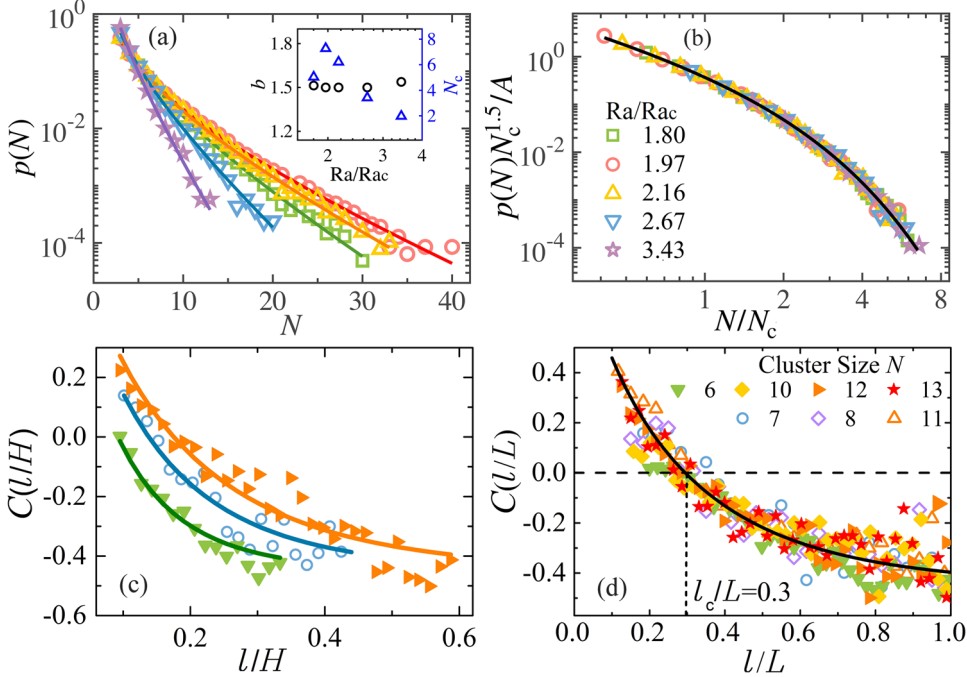

**Fig. 5 Statistical properties of vortex clusters and collective motion of vortices. a** Size distribution of vortex clusters. Solid curves represent fits to the experimental data $p(N) = AN^{-b}e^{-N/N_c}$. Results for $Ra = 3.0 \times 10^7$. Symbols are as in (**b**). Inset: the fitting parameters, i. e., the exponent $b$ (circles) and cutoff size $N_c$ (triangles) as functions of $Ra/Ra_c$. The prefactor $A$ is determined by the normalization relation: $\Sigma_N p(N) = 1$. **b** The rescaled data $p(N)N_c^{1.5}/A$ shown in a log-log frame as a function of $N/N_c$. All data collapse onto a master curve shown by the solid curve. See Supplementary Fig. 10 for a power-function compensated plot of $p(N)$. **c**, **d** Correlations of the vortex velocity fluctuation as a function of $l/H$ (**c**), and as a function of $l/L$ (**d**). Solid symbols: experimental data for $Ra = 3.0 \times 10^7$, $Ra/Ra_c = 1.97$. Open symbols: DNS data for $Ra = 2.0 \times 10^7$, $Ra/Ra_c = 2.26$. The solid curves represent stretched exponential functions, $C(l) = (1+a)e^{(-c_1 l)^{c_2}} - a$, fitted to the data (described in Supplementary Note 9). The vertical dashed line in (**d**) indicates the correlation length $l_c \approx 0.3L$ determined by the zero-crossing position of $C(l/L)$.

collectively. The power-law exponent, $b = 1.5 \pm 0.04$, is found to be independent of $Ra/Ra_c$ (inset of Fig. 5a), and falls within the range of previous theoretical predictions[40,44]. For large $N$ we find that $p(N)$ evolves into an exponential tail with the cutoff size $N_c$ varying with $Ra/Ra_c$ and reaching a maximum at $Ra/Ra_c = 1.97$, where the vorticity ratio $\gamma_\omega$ is maximum (see Fig. 3a). The rescaled data $p(N)N_c^{1.5}/A$ as a function of $N/N_c$ thus collapse onto a master curve as shown in Fig. 5b.

Dynamical systems consisting of clustered entities often exhibit scale-invariant, collective motions[12]. Here, we further analyze the spatial correlation function of vortex velocity fluctuations within a vortex cluster $C(l) = \sum_{ij}[\vec{u}'_i(\vec{r}_i + \vec{l}) \cdot \vec{u}'_j(\vec{r}_j)\delta(l - l_{ij})]/[C_0 \cdot \sum_{ij}\delta(l - l_{ij})]$, where $\vec{u}'_i = \vec{u}_i - \vec{V}$ is the relative vortex velocity with respect to the mean cluster velocity $\vec{V} = \sum_i \vec{u}_i/N$, $l_{ij}$ is the distance between the vortex pair $(i, j)$ and $C_0$ is a normalization constant. $\delta(l-l_{ij})$ is a Dirac function selecting pairs of vortices separated by distance $l$. Figure 5c shows that $C(l)$ decreases as the distance $l$ increases, with the decay length depending on the cluster size $N$. In Fig. 5d, we present the correlation function $C(l/L)$, scaled by the cluster length $L$, for clusters with various sizes. This rescaling leads to the converging of the data onto a single curve representing a stretched exponential function, which crosses zero at the correlation length $l_c \approx 0.3L$ for all cluster sizes $N$. Thus the correlated motions of the vortices are long-range and scale-free, i.e., there is no characteristic length scale here except the length $L$ of the cluster. We remark that the scattering of data points at large distances ($l \approx L$), owing to insufficient statistics, has negligible influence on the determination of $l_c$.

## Discussion
Our study has revealed the formation of large-scale coherent structures, in the form of vortex clusters, in rotating thermal convection. Within each cluster the lighter cyclones submit to collective motions dominated by the heavier anticyclones, exhibiting outward, inverse-centrifugation motion. We find that the size-distribution $p(N)$ of the vortex clusters can be well represented by a fractional power function with an exponential cutoff. The observed robust three-half power scaling of $p(N)$ for small $N$ (see Supplementary Fig. 9a) suggests that the theory of aggregation[40–42] apply to a broad range of grouping phenomena, and may provide predictions for the clustering dynamics of vortices in the present highly nonlinear, buoyancy-driven convection systems.

For large $N$ we find that $p(N)$ decays exponentially and the cutoff size $N_c$ is maximum when the centrifugal effect is dominant. Further investigations reveal that $N_c$ is proportional to the ratio of the vortex population density over the separation rate of the vortices from the clusters (detailed in Supplementary Fig. 9c), analogous to various biological systems[42,45]. We thus attribute the exponential decay of $p(N)$ to the separating and aggregating process of vortices between the clusters and the ambient flows, which maintains a statistically stable cluster-size distribution. As is shown in Supplementary Fig. 9b, the vortex separation rate reaches a minimum when the asymmetry of the vorticity fields is maximum. Thus when the interaction between adjacent vortices is dominated by the anticyclonic flows, the cluster structures possess the maximum stability against separation, leading to the largest characterized cluster size $N_c$.

Last, we have discovered that the self-organized vortices exhibit scale-free correlations of velocity fluctuations, with the

correlation length being approximately 30% of cluster length. This phenomenon of scale-invariant dynamics is analogous to the collective behavior observed widely in bird flocks, bacterial colonies[10,11], and in active matters[12]. The present study sheds light from a different angle on the phenomenon of collective motion and may have broad implications in the studies of soft condensed matter, fluid physics, and biological systems.

## Methods

**Experimental setup**. In the present study, we used a cylindrical cell mounted on a rapidly rotating table. Its bottom plate was made of 35 mm thick oxygen-free copper, heated from below by a uniformly distributed electric wire heater. Its top plate was a 5 mm thick sapphire disc, cooled from above through circulating coolant. Its sidewall, made of 3 mm thick Plexiglas, was protected against the ambient temperature fluctuations by an adiabatic shield that maintained a constant temperature. Temperature inhomogeneities over the top and bottom plates and the adiabatic shield were within one percent of $\Delta T$ (temperature difference between the top and bottom plates) during the experiment. The rotating axis of the table was adjusted to be accurately parallel to the gravity. The rotation was set in the clockwise direction with the rotation vector pointing downward (see Fig. 1). The convection cell was then leveled, using a cross-test level with a precision of 0.02 mm/m placed on the top surface of the top plate, to better than 0.001 rad. For flow visualization, a PIV system was installed on the co-rotating frame. A thin light-sheet powered by a solid-state laser illuminated the seed particles in a horizontal plane at a fluid height $z = H/4$ (Fig. 1i). Images of the particle were captured through the top sapphire window by a high-resolution camera ($2448 \times 2050$ pixels). Two-dimensional velocity fields were extracted by cross-correlating two consecutive particle images. Each velocity vector was calculated from interrogation windows ($32 \times 32$ pixels), with 50% overlap of neighboring sub-windows to ensure sufficient accuracy and resolution[46]. For each measurement, we took image sequences at a time interval of 0.5 s with a typical acquisition time of 2.5 h. Detailed experimental schemes of vortex identification and tracking are provided in Supplementary Note 2.

**Numerical method**. In the DNS we solved the three-dimensional Navier–Stokes equations with the Boussinesq approximation

$$\frac{D\mathbf{u}}{Dt} = -\nabla P + \left(\frac{Pr}{Ra}\right)^{1/2}\nabla^2\mathbf{u} + \theta\hat{\mathbf{z}}$$
$$+ \left(\frac{Pr}{RaEk^2}\right)^{1/2}\mathbf{u}\times\hat{\mathbf{z}} - \frac{2rFr}{d}\theta\hat{\mathbf{r}}, \tag{1}$$

$$\frac{D\theta}{Dt} = \frac{1}{(RaPr)^{1/2}}\nabla^2\theta, \tag{2}$$

$$\nabla \cdot \mathbf{u} = 0. \tag{3}$$

Here, $\mathbf{u}$ is the fluid velocity, $\theta$ and $P$ are the reduced temperature and pressure. The last two terms in the momentum equation (Eq. (1)) represent the Coriolis force and the centrifugal force, respectively. The equations were nondimensionalized using $L$, $\Delta T$, and the free-fall velocity $U_f = \sqrt{\alpha g\Delta TL}$. The simulations were performed in a cylindrical sample with an aspect ratio $\Gamma = 4$ and no-slip boundaries at all walls. Equations (1–3) were solved using a fully parallelized DNS code CUPS based on finite volume method with 4th order precision. To increase computational efficiency without any sacrifice in precision, we used a multiple-resolution strategy, in which the temperature equation was solved in a finer grid than the momentum one, allowing for a sufficient resolution to resolve the Batchelor and Kolmogorov length scales. The grid resolutions along radial, azimuthal and vertical directions were $140 \times 384 \times 160$ for the momentum and pressure fields, and $280 \times 768 \times 160$ for the temperature field. Staggered grids were used in the simulations, which allowed the grid cells corresponding to the three velocity components to be shifted by half a grid cell. Grids were refined near boundaries, so that boundary layers can be resolved. In addition, we considered the flow fields with $Fr = 0$, excluding the centrifugal effect.

## Data availability

The data that support the findings of this study are available within this article, its Supplementary Information, or from the corresponding authors upon request. Source data are provided with this paper.

## Code availability

The simulation codes that have been used to produce the numerical results of this study are available from the corresponding authors upon request when appropriate.

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

## Acknowledgements

This work is supported by the National Science Foundation of China under Grant nos. 11772235, 12072144 and 11572230, an NSFC/RGC Joint Research Grant nos. 11561161004 and N_CUHK437/15, and by the Hong Kong Research Grants Council under Grant nos. 14301115 and 14302317, and by Center for Computational Science and Engineering at Southern University of Science and Technology.

## Author contributions

J.-Q.Z. and K.-Q.X. conceived and designed the research. S.-S.D., J.-Q.S., and H.-Y.L. conducted the experiments. K.L.C. and G.-Y.D. conducted the numerical simulations. J.-Q.Z. and K.-Q.X. wrote the paper.

## Competing interests

The authors declare no competing interests.
