## [Peer Review File · Nature Communications]

Reviewers' Comments:

Reviewer #1:

Remarks to the Author:

RE: Inverse centrifugal effect induced by collective motion of vortices in rotating turbulent convection

In this paper the authors consider the motion of vortices in turbulent rotating convection. They combine experimental and numerical results and interpret them using theoretical models. The main finding - collective motion of vortex clusters where lighter (warmer) vortices move outward - is due to the asymmetry induced by centrifugation, a normally unwanted side-effect that is unavoidable in experiments. The paper displays a suite of precision experiments combined with simulations that explicitly involve centrifugal buoyancy to exactly mimic the experiments.

I commend the authors on their work that brings together several novel techniques in the analysis of vortex motion in rotating convection leading to nice results. However, I must come to the conclusion that the nice results alone do not warrant publication in Nature Communications. I detail my objections below.

1. Scope

The authors consider motion of vortical plumes in rotating convection. These vortices are the characteristic mode of convection near the onset point of convection, i.e. the thermal forcing is just above the minimal strength to induce fluid motion, where below the threshold a static fluid layer with diffusive heat transfer is observed. Applications in geophysics and astrophysics tend to display orders of magnitude stronger forcing, so that a true turbulent flow develops. The mention of the word "turbulent" in the title is misplaced at best. On top of that, the effect that the authors describe is induced by exaggerated centrifugation, a force that is vanishingly small in natural systems. The effect that the authors describe constitutes a small dynamical effect that experts in the field of rotating convection may have to deal with and are interested in, but the scope is not much broader than that.

2. Physical picture of the inner vortex structure

While reading the paper I repeatedly saw signs that could point at the authors having a wrong picture of the interior structure of the vortical plumes. It starts at the top of the first column on the second page: "These columnar vortices ... of the columnar vortices." (8 lines). From this text I infer that the authors believe that the velocity and vorticity of the columnar vortices is essentially independent of the vertical position: cyclones are warm and warm vortices are and remain cyclonic, while anticyclones are cold and cold vortices remain anticyclonic. This would be a naive application of the Taylor-Proudman theorem, which indeed would point at all three velocity components being independent of the vertical coordinate z . This picture is confirmed on page 3, 2nd column, about halfway: "It is the case ... by the background turbulence [27]." where I can only infer that the authors believe the anticyclones to be cold, to be strong (i.e. large magnitude of vorticity) near the top, and only gradually lose strength as they move down the fluid bulk. The correct physical picture of these vortices is that the vorticity changes sign near the mid-plane. Hot vortices are formed near the bottom plate, are cyclonic there. Then, their vorticity diminishes as they move up through the domain, vorticity changes sign and becomes negative as they approach the top plate, so that warm plumes are anticyclonic near the top plate. Likewise, cold plumes start off cyclonic near the top plate, gradually lose vorticity as they sink through the layer, and end up with anticyclonic vorticity near the bottom plate. This picture is already nicely illustrated by Chandrasekhar in his 1961 book; such flow structures are expected just above onset of convection. It forms the basis of the vortex models cited by the authors [22,24,26]. When density variations are present, it is wise to use the so-called thermal-wind balance for interpretation rather than the Taylor-Proudman theorem. In fact, horizontal density variations allow for vertical variations of velocity.

3. Validation of the results

The vortex picture discussed before has serious consequences for the theoretical model introduced by the authors. Their analysis is exclusively done near the bottom plate. In fact, the reported effect may turn out to exhibit the opposite direction near the top. A crucial step in the validation is

thus to consider a vertical position close to the top plate to repeat the analysis.

Additionally, quite some authors have discussed secondary circulations set up by centrifugal buoyancy in cylindrical rotating convection. The authors could measure the mean flow velocity (averaged over time and azimuthal angle) and quantify how that effect contributes to the observed drift. Is there a simpler origin for the observed behaviour?

4. Occurrence of distinctly different flow structures

A major result of the studies of the asymptotically reduced governing equations in the limit of rapid rotation (e.g., Refs. [26,28]) is the distinction of a succession of flow structures as a function of Ra/Ra_c . Just above onset, Ra/Ra_c is slightly larger than 1, one observes so-called cells, a state just like the authors present in their figure 4. When Ra/Ra_c becomes larger than (approximately) 3 the situation changes to the state of so-called convective Taylor columns, vortical cores surrounded by a characteristic sheath of oppositely signed vorticity. At even larger Ra/Ra_c these give way to so-called plumes, vortices that have lost the sheath and are no longer vertically coherent throughout the fluid layer. For a detailed description of these structures I refer to Ref. [28]. The authors should interpret their results in the framework of these flow structures; they have also been shown to be appropriate for systems with no-slip plates as in the currently studied flow problem (see Stellmach et al., Phys. Rev. Lett. 2014).

5. Applicability of the model for vortex motion

The authors present a plausible model for vortex motion under the effect of centrifugation. The resulting equation can be applied and fitted to the experimental and numerical results to satisfaction. However, the theoretical model introduces three free parameters, lending great liberty and flexibility in the fitting procedure. A matching fit does not guarantee a correct model. Are there other ways to add credibility to the model and the appropriateness of the values of the fit parameters? Can the numerical values be predicted from theory? Can the system be tuned by first excluding part of the dynamical effects?

6. Importance of centrifugal buoyancy

The authors have considered different levels of centrifugation, quantified by the Froude number Fr . If centrifugation is indeed an important dynamical effect, the parameters from the dispersion model should display a clear dependence on Fr . I know that it is next to impossible to vary Fr independent of the other dimensionless groups Ra , Ek in experiments, but in simulations it is straightforward.

Some minor remarks:

- Page 6, left column, the sentence part "...may provide predictions for the clustering dynamics of vortices in the present highly nonlinear, turbulent systems." : the parameter range in this paper could be called weakly turbulent at best; the most prominently discussed case at $Ra/Ra_c = 1.62$ is weakly nonlinear, certainly not turbulent.
- The Q criterion compares the magnitudes of the rotational and rate-of-strain parts of the velocity gradient tensor. In the reduced 2D formulation this reduces to the equation mentioned in the supplement. Vorticity can be related to both rotation and strain. The statement that Q reveals the strength of the vorticity field (caption figure 4) is incorrect.
- Eq. (13) of the supplement is dimensionally incorrect.
- Another three-parameter fit is done with Eq. (16) of the supplement. What is the physical interpretation of the three fit parameters? In particular the role of the variable a requires a better explanation; the correlation of velocity should vanish at large distances.

Reviewer #2:

Remarks to the Author:

Ding et al. describe the collective motion of vortices in the system of Coriolis-centrifugal convection. Rotating convection with centrifugal buoyancy included is still a relatively unexplored area of thermal convection research. The authors show that large-scale vortex clusters form through local interactions in dense vortex populations. These interactions lead - very counterintuitively - to something the authors appropriately coined "inverse centrifugal effect", namely, that the warm vortices move outwards despite the centrifugal buoyancy force being

directed inwards. The results are underpinned by a combination of laboratory experiments and numerical simulations, and the methodology and analyses are very carefully and rigorously explained in the supplementary material. The relevance of this study is not purely fluid dynamical but extends to many other fields, such as swarm and flocking behaviours found in biological systems. Typically collective motions are described within a statistical physics framework; thus, the here presented complementary fluid dynamical viewpoint can significantly enrich our understanding of many natural systems.

The study and analysis are unique, novel, scientifically sound, and of general interest. Hence, in my opinion, the manuscript is suitable for publication in Nature Communications.

However, before that, there are a few points that the authors should address:

The authors only present results for $z/H = 1/4$. However, the authors have also DNS data available. How does the behaviour change for other vertical positions, e.g. $z/H = 3/4$. Centrifugal buoyancy breaks the top-bottom symmetry - does it yield a noticeable effect?

Instead of using the approximate (asymptotic) critical $Ra_c = 8.7 Ek^{-4/3}$, it might be worth to calculate the exact Ra_c (Chandrasekhar, 1961), especially considering how high Ek is in the present study.

This may lead to a better collapse of the data shown in fig. 3 a.

The authors state on page 2 that the TP theorem provides a comprehensive description of the flow structures. That is not correct. TP technically only applies to steady or time-averaged flows. It is not directly applicable to non-linear and turbulent rotating convection. It only provides a zeroth-order explanation of the quasi-two-dimensionality of the flow, see, e.g., any of the many works by R. Ecke and co-workers, or K. Julien and co-workers.

Figure 3 (b, c) and the corresponding discussion: how do the results of the DNS with and without centrifugal buoyancy compare?

Page 7, Numerical method: Were the $Fr = 0$ simulations indeed conducted in Cartesian periodic boxes? This seems to contradict the main text. But would also explain any differences seen in figure 3.

A detailed data table of the laboratory and numerical set-ups and results would be beneficial, also in terms of reproducibility.

Minor points

Generally: Please use parentheses in the denominator of fractions for inline equations to avoid ambiguity.

Defining either acronyms or also using the flow regime numbers more consistently throughout the manuscript might improve readability a little.

Fig. 1, 2, 3: Use angular brackets, not less-than-signs.

Fig. 1: Could the authors indicate the vertical position also in the caption, and provide an indication of the flow regimes to make the figure self-contained?

Fig. 4: The inset covers the data, making it a separate figure seems justified.

Page 1: "densely populated vortices" is a bit misleading, the fluid is densely populated by vortices not the other way round, use, e.g., "dense vortex population" instead.

Page 1-2: The coherent vortices are only observed in moderate Pr fluids, $Pr \gtrsim 1$.

Page 2: Is the working fluid water? What is the maximum temperature difference? Are NOB effects to be expected.

Page 3, second paragraph: Define ζ .

Page 4, first paragraph: Calling the behaviour universal in Ra and Γ is a bit of a stretch with only one order of magnitude in Ra only two aspect ratios.

Page 7, eq. (1): big brackets.

References: There are a few "capital letter typos", e.g. rossby instead of Rossby, coriolis instead of Coriolis.

Page 1, right column, line 3: gives -> give
Page 2, second paragraph: measurement -> measurements
Page 3, line 5: rapid -> rapidly

Reviewer #3:

Remarks to the Author:

This is an outstanding paper. Rotating Rayleigh-Benard convection is a really important flow, with great geophysical relevance. To identify a new counter-intuitive flow regime (such as this inverse centrifugal effect) is a major breakthrough, and a major surprise which will be of broad interest. The presented research demonstrates the existence of this regime very convincingly, with clear comparison of experiments and numerics. The authors are to be further commended as they have also identified the key physical process, namely the aggregation of super-vortical structures. This is beautifully and convincingly done. The use of various sophisticated analysis tools is compelling, and interesting analogies are drawn with other complex systems prone to emergent collective behaviour. This work is indeed likely to influence thinking through the identification of this emergent intermediate regime. My only real criticism is cultural, and unlikely to carry much weight: I find it much easier to read more extended papers where everything is presented together, rather than the main paper with supplementary materials. I appreciate that such a structure is perhaps more appropriate for high impact journals, and in no way do I wish to imply that the main body of the paper is incoherent. It is very well-written and clear (and the figures are well-designed) I just have found a few trivial infelicities.

pg 1: "flocking of birds"

pg 3: "a theoretical model that consists of a set of Langevin equations"

pg 3: "A key question remaining to be answered"

Reviewer: 1

RE: Inverse centrifugal effect induced by collective motion of vortices in rotating turbulent convection

In this paper the authors consider the motion of vortices in turbulent rotating convection. They combine experimental and numerical results and interpret them using theoretical models. The main finding - collective motion of vortex clusters where lighter (warmer) vortices move outward - is due to the asymmetry induced by centrifugation, a normally unwanted side-effect that is unavoidable in experiments. The paper displays a suite of precision experiments combined with simulations that explicitly involve centrifugal buoyancy to exactly mimic the experiments.

I commend the authors on their work that brings together several novel techniques in the analysis of vortex motion in rotating convection leading to nice results. However, I must come to the conclusion that the nice results alone do not warrant publication in Nature Communications. I detail my objections below.

We thank the Referee for reviewing our manuscript. We are pleased to read that the Referee is of the view that our work brings together several novel techniques in the analysis of vortex motion in rotating convection. The comments/criticisms from the Referee are valuable and constructive for improving both the content and the presentation of our paper. Below are our responses (displayed in blue color) to each of the Referee's comments and questions (in black color). All changes made in the revised manuscript are displayed in red.

1. Scope

The authors consider motion of vortical plumes in rotating convection. These vortices are the characteristic mode of convection near the onset point of convection, i.e. the thermal forcing is just above the minimal strength to induce fluid motion, where below the threshold a static fluid layer with diffusive heat transfer is observed. Applications in geophysics and astrophysics tend to display orders of magnitude stronger forcing, so that a true turbulent flow develops. The mention of the word "turbulent" in the title is misplaced at best. On top of that, the effect that the authors describe is induced by exaggerated centrifugation, a force that is vanishingly small in natural systems. The effect that the authors describe constitutes a small dynamical effect that experts in the field of rotating convection may have to deal with and are interested in, but the scope is not much broader than that.

In response to the Referee's comment that the vortex motions we study are in characteristic mode of rotating convection near onset, we show in Fig. A1 in this reply (and in Supplementary Information (SI) Fig. 1), the parameter range of our experimental and numerical data in the phase diagram of $(Ek, Ra/Ra_c)$, and indicate the four states of various flow morphologies in rotating convection. The critical Rayleigh number Ra_c is calculated using the exact relation, $Ra_c = (8.70 - 9.63 \cdot Ek^{1/6}) \cdot Ek^{-4/3}$, as suggested by Referee #2 (thus Ra_c was underestimated in the previous version). Our data cover the range $(4.9 \times 10^{-6} \leq Ek \leq 2.7 \times 10^{-4}, 1.3 \leq Ra/Ra_c \leq 166)$, and thus we have studied flow fields *not only near the onset of convection, but also over a wide range spanning from cells, convective Taylor columns, plumes/geostrophic turbulence and finally to buoyancy-dominated convection*. Some of these flow regimes are relevant to important applications in geophysics and astrophysics. For instance, it is predicted that the convection flows in planetary cores as often organized into columnar vortices which are important for large-scale magnetic field generation (Olson et. al., *J. Geophys. Res.* 1999; Jones,

Annu. Rev. Fluid Mech. 2011). Geostrophic turbulence is another important convection state relevant to many large-scale flows in Atmospheric and Oceanic systems (Rhines, *Annu. Rev. Fluid Mech.* 1979; Vallis, *Atmospheric and Oceanic Fluid Dynamics*, 2006).

Since the inverse-centrifugal motions we study extensively in the paper occur mainly in the columnar state (although some are in cells and the plumes/GT states), we agree with the Referee that the phrase "turbulent convection" in the title may not best describe the present study (see our reply to the 4th question below). We have now changed the title to **"Inverse centrifugal effect induced by collective motion of vortices in rotating thermal convection"**. We appreciate this constructive suggestion from the Referee.

Fig. A1 Phase diagram of rotating convection and the parameter data of the present study. The parameter space (Ek , Ra/Ra_c) is divided into four convection states, i.e., Buoyancy-dominated, Plumes/Geostrophic Turbulence (GT), Columns and Cells, as suggested in the literature. Dotted red line ($Ra=3.4Ek^{-1.7}$): scaling of the boundary between buoyancy-dominated convection and geostrophic turbulence (Kunnen, *J. Turbul.* 2021; Lu et. al. arXiv: 2007.13279). Dash-dotted blue line ($Ra=1.18Ek^{-1.6}$): boundary between Plumes/GT and Columns for aspect ratio $\Gamma \geq 2$ (Lu et.al. arXiv: 2007.13279). Solid black line ($Ra/Ra_c=2$): boundary between Columns and Cells states (Julien et. al. *GAFD*, 2012; Aurnou et. al. *PEPI*, 2015; Kunnen, *J. Turbul.* 2021).

We respectfully disagree with the Referee's comment, "the effect that the authors describe is induced by exaggerated centrifugation, a force that is vanishingly small in natural systems". In the present work we consider a Froude-number range $0 \leq Fr \leq 0.27$ (see Supplementary Table 1 and 2). This is in fact a flow regime with modest centrifugal force, which is relevant to realistic flows in various natural and industrial systems as described below.

- (i) For mesocyclones in the atmosphere, such as tornadoes and hurricanes, the flow field are often dominated by the centrifugal force since the Froude number $Fr = \Omega^2 r / g$ can reach the order 1 (typical range of tornadic radius $50m \leq r \leq 100m$ and wind speed $50m/s \leq U \leq 85m/s$, see Tanamachi et. al. *Month. Wea. Rev.* 2007; Rotunno, *Annu. Rev. Fluid Mech.* 2013). Thus the centrifugal force is commonly incorporated in theoretical and numerical models to provide insights for the prominent flow structures in these mesocyclones (Rotunno, *Annu. Rev. Fluid Mech.* 2013; Montgomery and Smith, *Annu. Rev. Fluid Mech.* 2017).
- (ii) It has been reported that the centrifugation effect plays a significant role in determining the atmospheric flows on some giant gas planets. For instance, the Froude number of Jupiter's atmosphere is as large as 0.1 (Kaspi et.al. *Nature*, 2018). And the centrifugal

buoyance is regarded as an important term in modeling the convective flows on Jupiter (George et.al. *The Astrophysical Journal*, 2011; Kaspi et.al. *Nature*, 2018).

- (iii) In turbomachinery, thermal convection in a rapidly rotating cylindrical annulus occurs where the centrifugal acceleration can be orders of magnitude larger than the gravitational acceleration. In its operation period, the outer cylindrical surface is at a higher temperature than the inner one, leading to natural convection driven mainly by centrifugal buoyancy. Centrifugation-driven convection in rotating cavities has been used as a paradigmatic model to study the flow structures and heat transfer in turbine engines (see e.g. review papers by Bohn et.al. *ASME J. Turbomach.* 1995; Owen and Long, *ASME J. Turbomach.* 2015.)

In addition, with the progresses made in improved experimental techniques and numerical methods, centrifugation-driven convection has become a subject that receives considerable research interest in recent decades (e.g. Hart and Ohlsen *Phys. Fluids* 1999; Lopez and Marques, *J. Fluid Mech.* 2009; Horn and Aurnou, *Phys. Rev. Letts.* 2018; Jiang et. al. *Sci. Adv.* 2020). We believe that our present study of the rich vortex dynamics has made substantial contributions to the advances of this developing field.

Regarding the Referee's comment "The effect that the authors describe constitutes a small dynamical effect that experts in the field of rotating convection may have to deal with and are interested in, but the scope is not much broader than that", we first remark that the major discovery reported in the present study, i.e., self-aggregated vortices exhibit long-range, scale-free collective motion, is a striking phenomenon that has never been observed in a rotating fluid flow. Moreover, it is this collective motion of the vortices that leads to the inverse-centrifugal effect. Therefore, the inverse centrifugal motion we discover is a new phenomenon, rather than a small dynamical effect. It reveals an intrinsic dynamical feature of densely-populated vortices that is distinctively different from the behaviors of isolated vortices.

Many non-equilibrium systems in nature consisting large number of objects often exhibit self-organized collective motions. Examples include bird flocks, fish schools, bacteria swarms and clustering of active matters, etc. Through quantitative analysis, we provide concrete evidence that densely populated vortices in thermal convection exhibit similar behaviors, as manifested by the two defining features of collective motion: (i) a generic size-distribution $p(N)$ of vortex clusters, i.e., a fractional power function with an exponential cutoff, suggesting that the formation of the large-scale vortex clusters is a dynamical process of vortex aggregation; (ii) a universal correlation function $C(l/L)$ of velocity fluctuations between pairs of vortices, with the correlation length l proportional to the cluster size L , which implies that the collective motion of the vortices is scale-free. We expect that these discoveries should substantially advance our understanding of the vortex dynamics and interactions in rotating fluids, and bring to the broad community of fluid mechanics, statistical physics and biophysics new perspectives to the complex phenomena of grouping and collective motion.

We would like to further point out that the significance and broad interest of the present work have been highly appreciated by the other two Referees of our paper. For instance, Referee 2 has remarked in his/her report: "*The relevance of this study is not purely fluid dynamical but extends to many other fields.*" "*The study and analysis are unique, novel, scientifically sound, and of general interest*". And Referee 3 comments in his/her report: "*This is an outstanding paper.*" "*To identify a new counter-intuitive flow regime (such as this inverse centrifugal effect) is a major breakthrough, and a major surprise which will be of broad interest.*" "*The use of various sophisticated analysis tools is compelling, and interesting analogies are drawn with other complex systems prone to emergent collective behaviour*".

2. Physical picture of the inner vortex structure

While reading the paper I repeatedly saw signs that could point at the authors having a wrong picture of the interior structure of the vortical plumes. It starts at the top of the first column on the second page: "These columnar vortices ... of the columnar vortices." (8 lines). From this text I infer that the authors believe that the velocity and vorticity of the columnar vortices is essentially independent of the vertical position: cyclones are warm and warm vortices are and remain cyclonic, while anticyclones are cold and cold vortices remain anticyclonic. This would be a naive application of the Taylor-Proudman theorem, which indeed would point at all three velocity components being independent of the vertical coordinate z . This picture is confirmed on page 3, 2nd column, about halfway: "It is the case ... by the background turbulence [27]." where I can only infer that the authors believe the anticyclones to be cold, to be strong (i.e. large magnitude of vorticity) near the top, and only gradually lose strength as they move down the fluid bulk. The correct physical picture of these vortices is that the vorticity changes sign near the mid-plane. Hot vortices are formed near the bottom plate, are cyclonic there. Then, their vorticity diminishes as they move up through the domain, vorticity changes sign and becomes negative as they approach the top plate, so that warm plumes are anticyclonic near the top plate. Likewise, cold plumes start off cyclonic near the top plate, gradually lose vorticity as they sink through the layer, and end up with anticyclonic vorticity near the bottom plate. This picture is already nicely illustrated by Chandrasekhar in his 1961 book; such flow structures are expected just above onset of convection. It forms the basis of the vortex models cited by the authors [22,24,26]. When density variations are present, it is wise to use the so-called thermal-wind balance for interpretation rather than the Taylor-Proudman theorem. In fact, horizontal density variations allow for vertical variations of velocity.

We thank the Referee for commenting on the important problem of the inner vortex structure. We in fact do have the correct physical picture about the vortex structure. In the previous version of the manuscript, we have stated that "These columnar vortices are helical structures with their vorticity correlated to the temperature, i.e., cyclones are warm (upwelling) vortices while anticyclones are cold (downwelling) ones if observed from near the bottom boundary" (page 2, 1st paragraph), but probably not very clearly. In a previous publication (Shi et. al. *Phys. Rev. Fluids*, 5, 011501(R), 2020), we have examined the fine inner structures of the vortices in both the upper- and lower-half fluid layer, through high-resolution PIV measurements under similar experimental conditions. In that study we showed that in the lower half fluid layer cyclones (anticyclones) are upwelling (downwelling) vortices, but they change sign and become anticyclones (cyclones) as they cross the mid-plane (see Fig. 1 and discussions in page 2, last paragraph, of Shi et. al.). We apologize that we have not expressed ourselves clearly about this physical picture, which has led the Referee to think that we had an incorrect understanding about the vortex structure.

In the revised manuscript, we further demonstrate the three-dimensional vorticity fields in the insets of Fig. 3a. There our numerical data reveal clearly the vertical variation of the vorticity fields, and show that the vortex structures (in terms of their temperature fields) become strongly dependent on the fluid depth as the convection system changes from the rotation-dominated domain (left inset) to the buoyancy-dominated domain (right inset). This depth-dependence of the vortex structures, as the Referee has pointed out correctly, is owing to thermal-wind balance, i.e., horizontal temperature gradients allows for vertical variations of the vorticity fields.

We have now revised the relevant discussions in the manuscript. In page 2, first paragraph: "These columnar vortices are helical structures with either upward or downward flows.

Upwelling vortices rotate in the same direction as the system (cyclones) in the lower half of the fluid layer, and in the opposite direction (anticyclones) in the upper half, and vice versa for downwelling ones.” And “The theory of thermal wind balance, which relates the vertical variations of the fluid velocities with the horizontal temperature gradients [31], provides a general description of the flow structures of the columnar vortices.” In page 3, last paragraph: “It is the case because when observed at the lower half of the layer, anticyclones are downwelling vortices generated from the top boundary. They travel a longer distance to the measurement layer than the upwelling vortices (cyclones), and their momentum and vorticity have been largely dissipated by the background turbulence when reaching the measurement position in this flow regime [27].” And “With increasing Ω the up- and down-welling vortices evolve into columnar structures that are vertically antisymmetric in vorticity with respect to the mid-height plane [25, 26].”

3. Validation of the results.

The vortex picture discussed before has serious consequences for the theoretical model introduced by the authors. Their analysis is exclusively done near the bottom plate. In fact, the reported effect may turn out to exhibit the opposite direction near the top. A crucial step in the validation is thus to consider a vertical position close to the top plate to repeat the analysis.

We would like to first clarify that in our theoretical model we consider the vortices as inertial coherent structures. Their horizontal motions are determined by the volume averages over each vortex, of all the driving forces, including the centrifugal force, viscous damping, and the random forcing from the background turbulent flows (see Eq. 2 in the SI). In this sense, the centrifugal acceleration of each vortex is given by its volume-integrated density, but not dependent explicitly on the vertical variations of the vortex structure.

Following the Referee’s suggestion, we have studied and analysed vortex motions in vertical positions both near the top and bottom plates. Since the vortices exhibit horizontal motions quite differently in the strong-rotation and weak-rotation regimes, we discuss in below our results separately in these two regimes to fully validate our theoretical model. These discussions and the associated data figures are now added in the Supplementary Information of the paper.

(1) Vortex motion in the strongly rotating, inverse-centrifugal regime. In this regime where the vortices appear as columnar vortices (e.g. left inset in Fig. 3), their horizontal motions are similar at all fluid heights. We show in Fig. A2 below (see also Fig. 4 in the SI), the first moment $\langle r/r_0 \rangle_\xi$ and the second moment $\langle (r - \langle r \rangle_\xi)^2 \rangle_\xi$ of the vortex radial displacements in two vertical positions $z/H=0.2$ and $z/H=0.8$. One sees that for the two vertical positions the statistical results of both the first and the second moments are nearly identical. They reveal the outward motion of the cold vortices and the abnormal outward motion of the warm vortices. (Note that the statistical time for cold vortices are longer than the warm ones because they are relatively stronger and possess in average a longer lifetime.) The cold vortex motion observed in both vertical positions are found to be well described by our theoretical model (see SI 3). Panels (c) and (f) of Fig. A2 present respectively trajectories of a cold and a warm vortex, and show that the vortex trajectories measured at the two fluid heights for the same vortex indeed nearly match with each other.

Fig. A2 DNS data for the first and second moments of radial displacements of warm (a, b) and cold (d, e) vortices. Blue squares: data for $z=0.8H$. Red circles: data for $z=0.2H$. Individual vortex trajectories are shown in (c) for a warm vortex, and (f) for a cold vortex, obtained at the two heights (colours correspond to the same legends as in other panels). Results are for $Ra=2.0 \times 10^7$, $Ra/Ra_c=1.76$ and $Fr=0.174$.

(2) Vortex motion in the centrifugation-influenced regime with relatively slow rotations. In this flow regime the vortex structures are strongly dependent on the fluid height, the cold (warm) vortices forming at the top (bottom) boundary layer do not reach the opposite side of the fluid layer (see the right inset in Fig. 3 of the manuscript, and contrasting it with the left inset). For these reasons we observe less cold (warm) vortices in the lower (upper) fluid level in this regime. Figure A3 below (and also in SI Fig. 5) shows results of the first and second moments of the vortex radial displacements at two fluid depths. As expected the statistical data obtained for warm vortices at the upper fluid height ($z/H=0.8$) are less than that at the lower fluid level ($z/H=0.2$), and vice versa for the cold vortices. However, the vortices still exhibit clearly motions in the same direction in both the upper and the lower fluid layers. Over the short-time period where data of both types of vortices are available, their statistical moments are essentially the same, indicating the same physical mechanism that governs their horizontal motion.

Fig. A3 DNS data of the first and second moment of radial displacements of the vortices. Results for warm (a,b) and cold (c, d) vortices are shown. Blue squares: data for $z=0.8H$. Red circles: data for $z=0.2H$. Results for $Ra=2.0 \times 10^7$, $Ra/Ra_c=5.99$ and $Fr=0.029$.

Lastly, we find that for both the rapidly and slowly rotating cases, the statistical moments of the normal centrifugal motions of the vortices can be well predicted in our theoretical model (detailed in SI 3).

Additionally, quite some authors have discussed secondary circulations set up by centrifugal buoyancy in cylindrical rotating convection. The authors could measure the mean flow velocity (averaged over time and azimuthal angle) and quantify how that effect contributes to the observed drift. Is there a simpler origin for the observed behaviour?

By secondary circulations, we think that the Referee refers to a meridional circulation driven by the centrifugal buoyancy in which cool denser fluid is moving radially outward near the top boundary and warm lighter fluid moving inward near the bottom plate (e.g., Homsy and Hudson *J. Fluid Mech.* 1969; Hart and Ohlsen, *Phys. Fluids* 1999; Horn and Aurnou, *Phys. Rev. Fluids* 2019). Our data indeed reveal clearly the existence of this axisymmetric meridional circulation in flow regimes with $Ra/Ra_c \leq 10$. The radial component v_r of this circulation is found to be dependent on the radial position r and is in agreement with previous studies. Since $v_r(r, \theta)$ is influenced by the motions and the relative positions of the vortices, we take a mean-field approach. In Fig. A4 below we show the azimuthal- and time-averaged radial fluid velocity $\langle v_r \rangle$ obtained numerically as functions of Ra/Ra_c . Here $\langle v_r \rangle$ is measured at $r=d/4$, at a distance $z=Ek^{1/2}H$ from the top and bottom plates, where the magnitude of $\langle v_r \rangle$ is maximum. Results of $\langle v_r \rangle$ are compared with mean radial velocity $\langle u_r \rangle$ of both the warm (upwelling) and cold (downwelling) vortices. We see that overall $\langle u_r \rangle$ and $\langle v_r \rangle$ show very different dependences of Ra/Ra_c . The mean radial velocity of cold vortices $\langle u_r^c \rangle$ is larger than the meridional circulation velocity near the top boundary $\langle v_r^t \rangle$. When the centrifugal effect is dominant ($Ra/Ra_c \approx 2.0$), $\langle u_r^c \rangle$ becomes 4-5 times in magnitude greater than $\langle v_r^t \rangle$. This result rules out the scenario that the cold vortices are driven radially outward by the meridional circulation. For the radial velocity of warm vortices $\langle u_r^w \rangle$, we see that in the inverse-centrifugal regime (Regime II, $1.6 \leq Ra/Ra_c \leq 4.0$) $\langle u_r^w \rangle$ exhibits outward (inverse-centrifugal) motion as explained in the paper, which is opposite to the circulation direction $\langle v_r^b \rangle$ near the bottom boundary. Our data here thus provide direct evidence that the centrifugal circulation is not the origin for the observed horizontal motion of the vortices.

Fig. A4 The mean radial velocities of the vortices $\langle u_r \rangle$ (solid symbols, superscripts c and w are for cold and warm vortices respectively) and of the fluid circulations $\langle v_r \rangle$ near the top (t) and bottom (b) boundaries (open symbols), as function of Ra/Ra_c . Data are taken at radial position $r=d/4$. To represent the strength of the centrifugal circulations, the fluid velocities are determined at a distance $z=Ek^{1/2}H$ from the top and bottom plates, where the magnitude of $\langle v_r \rangle$ is maximum (see inset). The background color indicates the four different regimes of vortex motions. Inset: vertical profiles of $\langle v_r \rangle$ near the top and bottom boundaries. Results for $Ra/Ra_c=2.26$. Note that in regime IV, the vortices perform Brownian motion and therefore they do not exhibit a noticeable radial velocity.

Meanwhile, since the centrifugal circulation drives radial fluid motion inside the viscous boundary layers (BLs), we believe that this secondary flow may alter the viscous drag acting on the vortices during their horizontal motion. For instance, the outward circulation near the top boundary reduces (enhances) the viscous damping for the outward cold vortex (inward warm vortex) motion, and vice versa near the bottom boundary. In our theoretical model, the effect of viscous damping (mainly from the viscous BLs) for vortex motion, is parametrized by a relaxation timescale τ accounting for the damping effect, which is allowed to vary as functions of Ra/Ra_c independently for cyclones and anticyclones (SI Fig. 3). This modelling work has successfully interpreted the statistical properties of vortex motion as demonstrated in Fig. 2 of the manuscript.

4. Occurrence of distinctly different flow structures

A major result of the studies of the asymptotically reduced governing equations in the limit of rapid rotation (e.g., Refs. [26,28]) is the distinction of a succession of flow structures as a function of Ra/Ra_c . Just above onset, Ra/Ra_c is slightly larger than 1, one observes so-called cells, a state just like the authors present in their figure 4. When Ra/Ra_c becomes larger than (approximately) 3 the situation changes to the state of so-called convective Taylor columns, vortical cores surrounded by a characteristic sheath of oppositely signed vorticity. At even larger Ra/Ra_c these give way to so-called plumes, vortices that have lost the sheath and are no longer vertically coherent throughout the fluid layer. For a detailed description of these structures I refer to Ref. [28]. The authors should interpret their results in the framework of these flow structures; they have also been shown to be appropriate for systems with no-slip plates as in the currently studied flow problem (see Stellmach et al., Phys. Rev. Lett. 2014).

The Referee's description of the successive flow states that emerge with increasing Ra/Ra_c is certainly correct. We have shown in Fig. A1 in this reply (and Supplementary Fig. 1) the distinct convection states (including cells, convective Taylor columns, plumes and geostrophic turbulence) in the phase diagram of $(Ek, Ra/Ra_c)$, and noted the parameter range covered by the present study. In this context we find that in the centrifugation-influenced regime ($4 \leq Ra/Ra_c \leq 10$) the vortices are best described as plumes, while in the inverse-centrifugal regime ($1.6 \leq Ra/Ra_c \leq 4$) the vortices can be found to possess different types of structures (including plumes, columns and cells) depending on the relative strength of rotation over buoyancy.

We next discuss in detail concerning the Referee's comment that the flow transition between the cells and columns occurs approximately when $Ra/Ra_c=3$. As far as we are aware the only publication that suggested $Ra=3Ra_c$ to be the rough upper bound of weakly nonlinear convection regime (Ecke and Niemela, *PRL*, 2014). Meanwhile, we note in other literatures $Ra/Ra_c \approx 2$ is instead predicted by the asymptotic theory as the regime boundary (Julien et. al. *GAFD* 2012, i.e., Ref. [28] in the manuscript of the previous version). In that paper Julien et. al. reported that a boundary layer instability takes place at $RaEk^{4/3}=15$ (i.e., $Ra/Ra_c=1.72 \approx 2$, see Figs. 2 and 8 in their paper), which eventually leads to a regime transition from cells to columns. There are also a number of papers that show/use $Ra/Ra_c \approx 2$ as the transitional boundary (e.g. Nieves, et.al., *Phys. Fluids*, 2014; Aurnou et. al. *PEPI*, 2015; Cheng et. al. *Astrophys. Fluid Dyn.* 2018; Kunnen, *J. Turbul.* 2021).

In Fig. 4 of the manuscript we present structures of the vortices and their collective dynamics in the inverse-centrifugal regime. *We find that these vortices in fact possess columnar, rather than cellular structures*, since their vortical cores are each surrounded by a vortex sheath

with oppositely signed vorticity (see the measured vorticity profiles in SI Fig. 4). The example we show in Fig. 4 has $Ra/Ra_c=1.97$ that is comparable with the regime boundary ($Ra/Ra_c \approx 2$) between cells and columns predicted by the aforementioned asymptotic theory. We find that data with $1.6 \leq Ra/Ra_c \leq 4$ all exhibit similar inverse-centrifugal behavior.

We remark that the asymptotic approach, which has been very successful to capture the important physics of rapidly rotating RBC, has filtered out fast inertial waves to present only slow flow dynamics. The theory also eliminates the Ekman layers (Sprague et. al., *JFM*, 2006; Julien et. al. *GAFD*, 2012). As the Referee points out correctly, for systems with no-slip boundaries a subdominant boundary forcing must be complemented for better parameterizations (Stellmach et al., *PRL*, 2014). These facts may be responsible to the small discrepancy of flow-regime transitions observed theoretically and experimentally. Clearly, a precise theoretical classification of the parameter regimes for flow morphology still remains a challenge. In this respect, our present study, which combines results from laboratory experiments and direct numerical simulations, may provide new insights into the vortex structures and morphology in rotating convection.

5. Applicability of the model for vortex motion

The authors present a plausible model for vortex motion under the effect of centrifugation. The resulting equation can be applied and fitted to the experimental and numerical results to satisfaction. However, the theoretical model introduces three free parameters, lending great liberty and flexibility in the fitting procedure. A matching fit does not guarantee a correct model. Are there other ways to add credibility to the model and the appropriateness of the values of the fit parameters? Can the numerical values be predicted from theory? Can the system be tuned by first excluding part of the dynamical effects?

As we have discussed in the reply to the Referee's third question, we consider the vortices as inertial coherent structures in our theoretical model. The radial motion of the vortices is determined by volume averages of the centrifugal force, viscous damping, and the random forcing from the background turbulent flows. Since the Coriolis force acts on moving fluids both inside and outside the vortex structures, its overall effect of the radial vortex motion would be secondary compared to that from the centrifugal force. Therefore, our model contains three parameters, i. e., the momentum relaxation timescale τ , the diffusivity D and the fastest growth rate λ^* . They represent the strength of the viscous damping, the random forcing from background fluctuations and the centrifugal buoyancy, respectively. In order to provide constraints in fitting these parameters, we consider vortex motion in the slow rotating regime where the centrifugal force is negligible ($Fr \leq 0.0774$). In this regime we exclude the centrifugal effects and set the centrifugal coefficient ζ (and thus the fastest growth rate λ^*) to be zero. So the equation of motion for the vortices is reduced to the classical Langevin equation for Brownian motion, which contains only two free parameters (τ , D). We then performed data fitting for the mean-square-displacement given by Eq. (14) in the SI. Results of (τ , D) for the data set with $Ra=3 \times 10^7$ are shown as star symbols in Fig. 3 in the SI, which are seen to be in good accord with our results of three-parameter fitting. Thus in the slow rotating regime our model correctly asymptotes to the classical Langevin equation and interprets successfully the random Brownian motion of the vortices.

We note that in the randomly-diffusive regime ($Ra/Ra_c \approx 10$), the timescale of momentum relaxation τ for vortex motion could be approximated as the timescale of viscous diffusion for the vortices, $\tau \approx 2R^2/9\nu \approx 8s$. Here R is the characteristic size (i.e. the radius) of the vortices,

which is approximately 5 mm for $Ra/Ra_c \approx 10$ as found in our experiment; $\nu=6.7 \times 10^{-7} \text{ m}^2/\text{s}$ is the fluid viscosity. Such an order-of-magnitude estimate of τ appears in reasonable agreement with the predicted value from our model.

Finally, our model for vortex motion provides excellent predictions for several important statistical properties of the experimental data, including the linear profiles of the mean radial velocity $\langle u_r \rangle_\xi$ (Fig. 1, see discussions in the SI); and the super-diffusive (sub-diffusive) behaviour of anticyclones (cyclones) as demonstrated by the second moment of radial displacement $\langle (r - \langle r \rangle_\xi)^2 \rangle_\xi$ in Fig. 2.

Based on these evidences, we believe that our model captures the main underlying physics and provides satisfactory interpretations for the convective vortex motion under the centrifugal effect.

6. Importance of centrifugal buoyancy

The authors have considered different levels of centrifugation, quantified by the Froude number Fr . If centrifugation is indeed an important dynamical effect, the parameters from the dispersion model should display a clear dependence on Fr . I know that it is next to impossible to vary Fr independent of the other dimensionless groups Ra , Ek in experiments, but in simulations it is straightforward.

Prompted by the Referee's suggestion, we have performed new sets of direct numerical simulations varying Fr with other control parameters fixed as constants ($Ra=2.0 \times 10^7$, $Ra/Ra_c=4.57$). Figure A5 below shows results of (τ, λ^*, D) for $Fr=0.0$ (diamond), 0.02 (rightward triangle) and 0.08 (leftward triangle), which are compared with the original data set ($Fr=0.04$, plus). We see in Figs. A5(a) and (c) that results of (τ, D) remains largely unchanged if only the Froude number varies. This is predicted in our theoretical model: the momentum relaxation time scale τ is given by the viscous damping force and the diffusivity D is given by the strength of background fluctuation, and both are independent of the centrifugal coefficient.

Fig. A5 Parameters of the vortex motion as functions of Ra/Ra_c . (a) momentum relaxation timescale τ , (b) the fastest growth (slowest decay) rate λ^* for anticyclones (cyclones) and (c) diffusivity D (See also in Supplementary Fig. 3). Additional DNS results of anticyclones are shown for various Froude numbers, $Fr=0.0$ (blue diamond), $Fr=0.02$ (yellow rightward triangle) and $Fr=0.08$ (green leftward triangle), with $Ra=2.0 \times 10^7$, $Ra/Ra_c=4.57$ hold at constant. Inset in (b): λ^* as a function of Fr for $Ra/Ra_c=4.57$ (results are for anticyclones). Open squares: DNS data with $Ra=2.0 \times 10^7$; red circle: experimental data with $Ra=2.0 \times 10^7$; blue triangle: experimental data with $Ra=3.0 \times 10^7$.

However, the new data with artificial Froude numbers indicate that the fastest growth rate λ^* is sensitive to Fr . For the given parameter settings of $(Ra, Ra/Ra_c)$ we find in Fig. A5(b) that

λ^* increases monotonically with increasing Fr (seen as well in the inset), which is expected in our model. The inset of Fig. A5(b) also indicates that for a given Ra/Ra_c , λ^* is greater for a data set with a higher Ra. This may be understood using our model. In the model, the fastest growth rate is given by $\lambda^* = (1/4\tau^2 + \alpha\delta T\Omega^2)^{1/2} - 1/2\tau \approx \tau\alpha\delta T\Omega^2$. Here we have used the approximation $1/4\tau^2 \gg \alpha\delta T\Omega^2$, according to the known values of τ and λ^* as shown in Fig. A5. For the data set with a higher Ra but a fixed Ra/Ra_c , the Ek number is relatively lower and thus the rotation rate Ω is larger, this leads to a greater value of λ^* as observed in Figure A5(b).

Some minor remarks:

- Page 6, left column, the sentence part "...may provide predictions for the clustering dynamics of vortices in the present highly nonlinear, turbulent systems.": the parameter range in this paper could be called weakly turbulent at best; the most prominently discussed case at $Ra/Ra_c = 1.62$ is weakly nonlinear, certainly not turbulent.

We agree with the Referee's remark and have revised the manuscript: "...may provide predictions for the clustering dynamics of vortices in the present nonlinear, buoyancy-driven convection systems."

- The Q criterion compares the magnitudes of the rotational and rate-of-strain parts of the velocity gradient tensor. In the reduced 2D formulation this reduces to the equation mentioned in the supplement. Vorticity can be related to both rotation and strain. The statement that Q reveals the strength of the vorticity field (caption figure 4) is incorrect.

We thank the Referee for pointing this out. In the revised manuscript, we have revised the pertaining sentence in the caption of figure 4 as: "The background coloration represents distribution of the quantity Q/Q_{std} ."

- Eq. (13) of the supplement is dimensionally incorrect.

We find that there was an extra factor of t on the right-hand-side of Eq. (13), which was left there erroneously. We have now corrected this typo in the equation. We thank the Referee for pointing out this mistake.

- Another three-parameter fit is done with Eq. (16) of the supplement. What is the physical interpretation of the three fit parameters? In particular the role of the variable a requires a better explanation; the correlation of velocity should vanish at large distances.

The Referee is correct that the correlation function $C(l)$ of vortex velocity fluctuation contains three parameters (a , c_1 , c_2) (see Eq. 16 of the supplement). These three parameters characterize the several important features of $C(l)$ we discover when the vortices exhibit intriguing collective motion. (i) First of all, $C(l)$ takes the value of unity at zero vortex separation $l=0$ and decreases with increasing l , but remains above zero for $l \leq l_0$, indicating that the velocity $u' = u - \langle u \rangle$ of each vortex relative to the ensemble average of all vortices in the same cluster is positively correlated to each other at small distance. (ii) $C(l)$ crosses zero at the correlation length ($l=l_0$) and becomes negative when $l \geq l_0$. This suggests that the velocity fluctuations of the vortices become negatively correlated at large distance. Note that owing to

the definition of a vortex cluster (see discussions of Fig. 4 in the paper), *the motions of two vortices separated by the largest possible distance within a cluster ($l=L$) still remain correlated, and there we find $C(l=L)$ remains negative but non-zero.* These are the intrinsic statistical properties of collective motion observed as well in many natural systems (e.g. Cavagna et al. *Proc. Natl. Acad. Sci.* 2010; Chen et. al. *Phys. Rev. Lett.* 2012; Bialek et. al. *Proc. Natl. Acad. Sci.* 2012; Huepe, et. al. *J Stat. Phys.* 2015).

In order to represent these features of $C(l)$, several analytical functional forms have been proposed in previous studies. In the present work we use a stretched exponential function, $C(l) = (1 + a)e^{(-c_1 l)^{c_2}} - a$, as given in Eq. 16 in SI (see also Chen et. al. *Phys. Rev. Lett.* 2012). In this functional form, the scale factor $c_1=1/(0.245L)$ reveals the scale-invariant properties of $C(l)$, i. e., $C(l/L)$ is independent of the cluster size L and $C(l=0.3L) \cong 0$. The exponent $c_2=0.85$ indicates that for large separation l the correlation function $C(l)$ decreases slower than an exponential function. Finally, the residue coefficient $a=0.45$ represents the finite (negative) value of $C(l)$ for large vortex separation.

Finally, we thank the referee once again for the critical and constructive comments and suggestions, which have helped improving both the content and presentation of our manuscript. We hope with these improvements our manuscript can now be recommended for publication in Nature Communications.

Reviewer: 2

Ding et al. describe the collective motion of vortices in the system of Coriolis-centrifugal convection. Rotating convection with centrifugal buoyancy included is still a relatively unexplored area of thermal convection research. The authors show that large-scale vortex clusters form through local interactions in dense vortex populations. These interactions lead – very counterintuitively - to something the authors appropriately coined "inverse centrifugal effect", namely, that the warm vortices move outwards despite the centrifugal buoyancy force being directed inwards. The results are underpinned by a combination of laboratory experiments and numerical simulations, and the methodology and analyses are very carefully and rigorously explained in the supplementary material. The relevance of this study is not purely fluid dynamical but extends to many other fields, such as swarm and flocking behaviours found in biological systems. Typically collective motions are described within a statistical physics framework; thus, the here presented complementary fluid dynamical viewpoint can significantly enrich our understanding of many natural systems.

The study and analysis are unique, novel, scientifically sound, and of general interest. Hence, in my opinion, the manuscript is suitable for publication in Nature Communications.

We are pleased to read that the Referee is of the view that the study and analysis of the present work are unique, novel, scientifically sound, and of general interest. Our study is not purely fluid dynamical but extends to many other fields, such as swarm and flocking behaviours found in biological systems. The presented complementary fluid dynamical viewpoint can significantly enrich our understanding of many natural systems. We thank the Referee for recommending the manuscript for publication in Nature Communications.

Below are our responses (displayed in blue color) to each of the comments and questions (in black color). All changes made in the revised manuscript are displayed in red.

However, before that, there are a few points that the authors should address:

The authors only present results for $z/H = 1/4$. However, the authors have also DNS data available. How does the behaviour change for other vertical positions, e.g. $z/H = 3/4$. Centrifugal buoyancy breaks the top-bottom symmetry - does it yield a noticeable effect?

We thank the referee for this insightful comment. Following the Referee's suggestion, we have studied and analysed vortex motions in vertical positions both near the top and bottom plates. Since the vortices exhibit horizontal motions quite differently in the strong-rotation and weak-rotation regimes, we discuss in below our results separately in these two regimes to fully validate our theoretical model. These discussions and the associated data figures are now added in the supplement of the paper.

(1) Vortex motion in the strongly rotating, inverse-centrifugal regime.

In this regime where the vortices appear as columnar vortices (e.g. left inset in Fig. 3), their horizontal motions are similar at all fluid heights. We show in Fig. B1 below (see also Fig. 4 in the Supplementary Information (SI)), the first moment $\langle r/r_0 \rangle_\xi$ and the second moment $\langle (r - \langle r \rangle_\xi)^2 \rangle_\xi$ of the vortex radial displacements in two vertical positions $z/H=0.2$ and $z/H=0.8$. One sees that for the two vertical positions the statistical results of both the first and the second moments are nearly identical. They reveal the outward motion of the cold vortices and the abnormal outward motion of the warm vortices. (Note that the statistical average time for cold

Fig. B1 DNS data for the first and second moments of radial displacements of warm (a, b) and cold (d, e) vortices. Blue squares: data for $z=0.8H$. Red circles: data for $z=0.2H$. Individual vortex trajectories are shown in (c) for a warm vortex, and (f) for a cold vortex. Results for $Ra=2.0 \times 10^7$, $Ra/Ra_c=1.76$ and $Fr=0.174$.

vortices are longer than the warm ones because they are relatively stronger and possess in average a longer lifetime.) The cold vortex motion observed in the two vertical positions are found to be well described by our theoretical model (see SI 3). Panels (c) and (f) of Fig. B1 present respectively trajectories of a cold and a warm vortex, and show that the vortex trajectories measured at the two fluid heights for the same vortex indeed match approximately with each other.

We remark that for this parameter range ($Fr=0.174$) the centrifugal buoyancy is known to break the top-bottom symmetry of the temperature field (Horn and Aurnou, *Phys. Rev. Fluids*, 2019). However, our data suggest that *this asymmetry does not lead to noticeable effect on the horizontal motion of the vortices.*

Fig. B2 DNS data for the first and second moment of radial displacements of the vortices. Results for warm (a,b) and cold (c, d) vortices are shown. Blue squares: data for $z=0.8H$. Red circles: data for $z=0.2H$. Results for $Ra=2.0 \times 10^7$, $Ra/Ra_c=5.99$ and $Fr=0.029$.

(2) Vortex motion in the centrifugation-influenced regime with relatively slow rotations.

In this flow regime the vortex structures are strongly dependent on the fluid height, the cold (warm) vortices forming at the top (bottom) boundary layer do not reach the opposite side of the fluid layer (see the right inset in Fig. 3 of the manuscript, and contrasting it with the left

inset). For these reasons we observe less cold (warm) vortices in the lower (upper) fluid level in this regime. Figure B2 below (and also in SI Fig. 5) shows results of the first and second moments of the vortex radial displacements at the two fluid depths. As expected the statistical data obtained for warm vortices at the upper fluid height ($z/H=0.8$) are less than that at the lower fluid level ($z/H=0.2$), and vice versa for the cold vortices. However, the vortices still clearly exhibit motions in the same direction at both the upper and lower fluid layers. In the short-time domain where data of both types of vortices are available, their statistical moments overlap with each other, indicating the same physical mechanism that governs their horizontal motion.

Lastly, we find that for both the rapidly and slowly rotating cases, the statistical moments of the normal-centrifugal vortex motions can be well predicted in our theoretical model (detailed in SI 3).

Instead of using the approximate (asymptotic) critical $Ra_c = 8.7 Ek^{-4/3}$, it might be worth to calculate the exact Ra_c (Chandrasekhar, 1961), especially considering how high Ek is in the present study. This may lead to a better collapse of the data shown in fig. 3 a.

We thank the Referee for this helpful comment. We have recalculated critical Rayleigh number using the exact relation, $Ra_c = (8.70 - 9.63 \cdot Ek^{1/6}) \cdot Ek^{-4/3}$ and updated its values in the revised manuscript accordingly. It indeed leads to a slightly better collapse of the data shown in Fig. 3a, particularly in the low Ra/Ra_c regime. With this revision the figure presents more accurately the various convective flow states explored in the current study

The authors state on page 2 that the TP theorem provides a comprehensive description of the flow structures. That is not correct. TP technically only applies to steady or time-averaged flows. It is not directly applicable to non-linear and turbulent rotating convection. It only provides a zeroth-order explanation of the quasi-two-dimensionality of the flow, see, e.g., any of the many works by R. Ecke and co-workers, or K. Julien and co-workers.

We agree with the Referee that our previous interpretation of the flow structures under finite rotational strength in terms of the TP theorem is incorrect. The TP theorem indeed provides only a zeroth-order description for rotation-dominant, quasi-two-dimensional flows. In the present rotating RBC system, horizontal temperature gradient exists due to the thermal forcing. It causes vertical velocity gradients in the bulk flows and thus the depth-dependence of the vortex structures (the thermal wind balance).

We have now revised the relevant discussions in the manuscript. In page 2, first paragraph: “These columnar vortices are helical structures with either upward or downward flows. Upwelling vortices rotate in the same direction as the system (cyclones) in the lower half of the fluid layer, and in the opposite direction (anticyclones) in the upper half, and vice versa for downwelling ones.” And “The theory of thermal wind balance, which relates the vertical variations of the fluid velocities with the horizontal temperature gradients [31], provides a general description of the flow structures of the columnar vortices.” In page 3, last paragraph: “It is the case because when observed at the lower half of the layer, anticyclones are downwelling vortices generated from the top boundary. They travel a longer distance to the measurement layer than the upwelling vortices (cyclones), and their momentum and vorticity have been largely dissipated by the background turbulence when reaching the measurement position in this flow regime [27].” And “With increasing Ω the up- and down-welling vortices

evolve into columnar structures that are vertically antisymmetric in vorticity with respect to the mid-height plane [25, 26].”

Figure 3 (b, c) and the corresponding discussion: how do the results of the DNS with and without centrifugal buoyancy compare?

In response to the Referee’s question, we have performed new numerical simulations with the same parameter/boundary condition settings as those in Fig. 3 (b, c) of the manuscript, except that the centrifugal buoyancy is excluded. In Fig. B3(a) below we show that when the centrifugal buoyancy is absent the background fluid temperature T_c is close to the arithmetic mean of the top and bottom boundary temperature (T_c is slightly larger than 0 because the data are taken at the lower half of the fluid layer, $z=0.2H$). Moreover, in the bulk region ($r \leq 0.4d$) the mean temperatures of cold and warm vortices locate nearly symmetrically on the lower and upper sides of T_c , which contrasts strongly with the case when centrifugal buoyancy is present. Figure B3 (b) shows that the ratios of vortex temperature (γ_T) and vorticity (γ_ω) are identical and close to unity in the bulk (they are slightly less than 1.0 because again the data are taken at $z=0.2H$ closer to bottom), again, in strong contrast to the case of non-zero centrifugal buoyancy. Distribution of the temperature anomaly $\delta T/\Delta T$ at the fluid depth $z=0.2H$ are shown in Fig. B3(c). In the near-sidewall region ($0.4d \leq r \leq 0.5d$) we see large fluctuations in the temperature fields owing to the perturbations of the boundary flows (e.g. Zhang et.al. *Phys. Rev. Letts.*, 2020).

Fig. B3 Numerical results with centrifugal buoyancy turned-off. (a) Radial profiles of the mean temperatures $\langle T - T_m \rangle / \Delta T$ for cyclones (red circles), anticyclones (blue triangles) and the background fluid T_c (solid line). The length of the dashed lines indicates the temperature difference δT between the cyclones (anticyclones) and the background fluid. (b) Radial profiles of $\gamma_\omega = |\langle \omega_a \rangle / \langle \omega_c \rangle|$ and $\gamma_T = |\langle \delta T_a \rangle / \langle \delta T_c \rangle|$. (c) Distribution of the temperature anomaly $\delta T/\Delta T$. The green dashed circle denotes a radial position $r/d=0.45$ where temperature fluctuations due to the boundary flows are dominant. DNS data obtained at $z=0.2H$ for $Ra=2.0 \times 10^7$ and $Ra/Ra_c=2.26$.

Overall, our DNS data with the centrifugal force excluded reveal a relatively symmetric flow field in the bulk region. These data are supportive of our interpretation that the centrifugal buoyancy breaks the symmetry of the temperature and vorticity fields, a central integration for the inverse-centrifugal vortex motion reported in the manuscript. We have added Fig. B3 and the above discussions in the SI of the paper.

Page 7, Numerical method: Were the $Fr = 0$ simulations indeed conducted in Cartesian periodic boxes? This seems to contradict the main text. But would also explain any differences seen in figure 3.

The Referee is correct that we presented in the previous version of Fig. 3a the $Fr=0$ DNS data with Cartesian periodic boundary conditions, which were compared to the data in cylindrical cells with the centrifugal buoyancy included.

Fig. B4 (a) The vorticity ratio $\gamma_\omega = |\langle \omega_a \rangle| / |\langle \omega_c \rangle|$ of the anticyclones over the cyclones as a function of Ra/Ra_c . Data from DNS with non-slip cylindrical boundary including (excluding) the centrifugal force are shown as pluses (open diamonds) for $\Gamma=4.0$. Data from DNS in periodic boxes excluding the centrifugal force are shown in open circles for $\Gamma=2.0$.

Prompted by the Referee's question we have made new simulations with $Fr=0$ but in a cylindrical fluid domain with non-slip boundaries. In these simulations we use $Ra=2.0 \times 10^7$ and $\Gamma=4.0$, the same parameters used in previous simulations that included the centrifugal force. In Fig B4 (and Fig. 3a of the revised manuscript), these new DNS data (which replace the old data with Cartesian periodic boundaries in the manuscript) show the same trend of $\gamma_\omega(Ra/Ra_c)$ when the centrifugal force is absent. Starting from approximately the value of 0.6 in the slow rotating regime, the vorticity ratio γ_ω increases monotonically with decreasing Ra/Ra_c , and approaches unity in the limit of rapid rotations. We thus conclude that it is the centrifugal buoyancy, rather than the non-slip boundaries or geometrical effects, that breaks the symmetry of the vorticity fields.

A detailed data table of the laboratory and numerical set-ups and results would be beneficial, also in terms of reproducibility.

We thank the Referee for this constructive comment. We have now provided in Supplementary Table 1 and 2 of the revised manuscript, all parameter values of (Ra , Ek , Ra/Ra_c , Fr and Γ) used in our laboratory and numerical studies. We also show in Supplementary Fig. 1 the parameter range covered in our study in the phase diagram of (Ek , Ra/Ra_c).

We have organized the experimental and numerical data presented in the main figures of the manuscript into a single Excel file ("Source Data.xlsx"), with data for each figure provided in a separate sheet. This data file is provided with the paper.

In order to further ensure the reproducibility of the present work, we have described in great detail in the "Methods" section, the experimental methods, accuracy, and the numerical schemes used in our study:

"Temperature inhomogeneities over the top and bottom plates and the adiabatic shield were

within one percent of ΔT (temperature difference between the top and bottom plates) during the experiment. The rotating axis of the table was adjusted to be accurately parallel to the gravity. The convection cell was then levelled, using a cross-test level with a precision of 0.02 mm/meter placed on the top surface of the top-plate, to better than 0.001 rad.”

“The grid resolutions along radial, azimuthal and vertical directions were $140 \times 384 \times 160$ for the momentum and pressure fields, and $280 \times 768 \times 160$ for the temperature field. Staggered grids were used in the simulations, which allowed the grid cells corresponding to the three velocity components to be shifted by half a grid cell. Details of the multiple-resolution algorithm can be found in Ref. [34]. Grids were refined near boundaries, so that boundary layers can be resolved.”

Finally, we have now included both the “Data Availability” and the “Code Availability” sections in the manuscript to inform the readers that these data sources are available from the corresponding authors upon request.

Minor points:

Generally: Please use parentheses in the denominator of fractions for inline equations to avoid ambiguity.

We have used parentheses in the denominator of fractions for all inline equations in the revised manuscript.

Defining either acronyms or also using the flow regime numbers more consistently throughout the manuscript might improve readability a little.

In the revised version we now use the flow regime numbers (Roman numerals) consistently throughout the manuscript.

Fig. 1, 2, 3: Use angular brackets, not less-than-signs.

We have fixed these formats in the axis labels of Figs. 1,2,3.

Fig. 1: Could the authors indicate the vertical position also in the caption, and provide an indication of the flow regimes to make the figure self-contained?

We thank the Referee for these suggestions. We have showed the vertical position of the measured fluid layer (position of the laser sheet) in the schematic drawing of Fig. 1(i), and explained it in the caption: “A laser sheet illuminates a rotating Rayleigh-Bénard convection cell filled with water and seeded with tracer particles at a fluid height $z=H/4$.”

We also provide information of the flow regimes in the caption:“(e-h) Radial profiles $\langle u_r(r) \rangle_\xi$ of cyclones and anticyclones. Data for $Ra=2.0 \times 10^7$ and from left to right, $Ra/Ra_c=20.9, 4.57, 2.26, 1.43$, corresponding to the four flow regimes (I), (II), (III) and (IV), respectively (see text for discussions).”

Fig. 4: The inset covers the data, making it a separate figure seems justified.

In viewing that there is not extra important information of vortex clustering in the bottom right corner of Fig.4 (overlaid by the inset), and for the purpose of better representation, we prefer to retain Fig. 4 as it is in the manuscript.

Page 1: "densely populated vortices" is a bit misleading, the fluid is densely populated by vortices not the other way round, use, e.g., "dense vortex population" instead.

We thank the Referee for this comment. We have now revised the manuscript as follows: "...in rapidly rotating turbulent flows these vortices become densely distributed".

Page 1-2: The coherent vortices are only observed in moderate Pr fluids, $Pr \gtrsim 1$.

We have now revised the manuscript as follows: "Recent studies report that for rapidly rotating RBC in moderate Prandtl-number fluids, the convective flows are organized by the Coriolis force into coherent columnar vortices."

Page 2: Is the working fluid water? What is the maximum temperature difference? Are NOB effects to be expected.

We use deionized water as the working fluid. The maximum temperature difference ΔT applied in the experiment is 6.46K (corresponding to the case for $H=63.0$ mm and $Ra=6.0 \times 10^7$). Thus the thermal expansion factor $\alpha \Delta T$ does not exceed 2.5×10^{-3} , and the NOB effects are negligible (see e. g. Horn and Shishkina, *Phys. Fluids*, 2014).

Page 3, second paragraph: Define ζ .

We have added the following sentence in the caption of Figure 1 (where ζ first appears) a definition of ζ : "Here ζ denotes individual vortex trajectory and $\langle \zeta \rangle$ a trajectory-ensemble average."

Page 4, first paragraph: Calling the behaviour universal in Ra and Gamma is a bit of a stretch with only one order of magnitude in Ra only two aspect ratios.

We have revised this sentence in the manuscript: "Remarkably, we observe that $\gamma_\omega(Ra/Ra_c)$ is independent of Ra and Γ over the parameter range studied."

Page 7, eq. (1): big brackets.

We now use big brackets in Eq. (1).

References: There are a few "capital letter typos", e.g. rossby instead of Rossby, coriolis instead of Coriolis.

Page 1, right column, line 3: gives -> give

Page 2, second paragraph: measurement -> measurements

Page 3, line 5: rapid -> rapidly

These typos have now been fixed in the revised manuscript.

Finally, we thank the referee once again for the critical and constructive comments. These have helped improving both the content and presentation of our manuscript.

Reviewer: 3

This is an outstanding paper. Rotating Rayleigh-Benard convection is a really important flow, with great geophysical relevance. To identify a new counter-intuitive flow regime (such as this inverse centrifugal effect) is a major breakthrough, and a major surprise which will be of broad interest. The presented research demonstrates the existence of this regime very convincingly, with clear comparison of experiments and numerics. The authors are to be further commended as they have also identified the key physical process, namely the aggregation of super-vortical structures. This is beautifully and convincingly done. The use of various sophisticated analysis tools is compelling, and interesting analogies are drawn with other complex systems prone to emergent collective behaviour. This work is indeed likely to influence thinking through the identification of this emergent intermediate regime. My only real criticism is cultural, and unlikely to carry much weight: I find it much easier to read more extended papers where everything is presented together, rather than the main paper with supplementary materials. I appreciate that such a structure is perhaps more appropriate for high impact journals, and in no way do I wish to imply that the main body of the paper is incoherent. It is very well-written and clear (and the figures are well-designed) I just have found a few trivial infelicities.

We thank the Referee for reviewing our manuscript carefully. We highly appreciate your remark that to identify a new counter-intuitive flow regime (such as the inverse centrifugal effect) is a major breakthrough, and a major surprise that is of broad interest. We are pleased that the Referee finds that “the use of various sophisticated analysis tools is compelling, and interesting analogies are drawn with other complex systems prone to emergent collective behaviour. This work is indeed likely to influence thinking through the identification of this emergent intermediate regime.” We agree with the Referee that it is for the purpose of focused discussions of the main important results in the main paper that we have organized the detailed supporting materials in the supplement.

Please find below the revisions we made in the manuscript (displayed in red color) in response to the Referee’s comments.

pg 1: "flocking of birds"

We have corrected this phrase as “bird flocks” in the revised manuscript.

pg 3: "a theoretical model that consists of a set of Langevin equations"

We have corrected this sentence as: “We formulate a theoretical model consisting Langevin-type equations that incorporate the centrifugal force.”

pg 3: "A key question remaining to be answered"

We have corrected this sentence as: “A key question is then what sets the anomalous vortex motion?”

Reviewers' Comments:

Reviewer #1:

Remarks to the Author:

RE: Inverse centrifugal effect induced by collective motion of vortices in rotating thermal convection
(Resubmission)

In this resubmission (and rebuttal) the authors have provided clear and compelling answers to my points pertaining to the original submission. I want to briefly reply to some answers of the authors to these points:

1. Scope

- I thank the authors for reconsidering the use of "turbulence". The main message of the paper is for a not-quite-turbulent flow state.
- The authors propose compelling examples of natural and technological flows where this study is directly relevant. I agree that my initial assessment was too restrictive. However, the first example of mesocyclones is not readily applicable; these flow structures set up their own centrifugation to act in the force balance (cyclostrophic balance). Centrifugation is here not related to outside effects (i.e. rotation of the Earth) and does not contribute to the displacement/trajectory of mesocyclones.
- On that note, why are some of these examples not briefly mentioned in the paper?

2. Physical picture of the inner vortex structure & 3. Validation of the results

Thank you for these clarifications and validations! These essential validation steps show that the main message of the paper is not a more basic effect of centrifugation in setting up a meridional circulation inside the cylinder. It is compelling to see that typical vortex drift velocities are significantly larger than the mean flow velocity of the centrifugation-induced meridional circulation.

I am pleased with how the authors have answered my points. This convinced me of the validity of the interpretations of the results. It is important to include the additional validation steps (discussed in the rebuttal) in the supplement. I can now recommend publication. I request the authors to consider my suggestion in item 1.

Reviewer #2:

Remarks to the Author:

The authors have addressed all of my previous concerns.

I was very pleased to see that the authors confirmed their results in cylindrical DNS instead of only Cartesian geometries. I also appreciated the additional analysis in the upper part of the fluid domain.

These newly added results make the paper even stronger.

I fully recommend the publication of this excellent work in Nature Communications.

The authors may however still want to consider the following minor remarks:

Is the rotation in the experiments in the clockwise direction? This seems to be the case based on figure 1 (e). If so, it should be made explicit somewhere since most studies are anticlockwise by default because Earth rotates anticlockwise. Also, the DNS are stated to be anticlockwise.

A suggestion regarding figure 3 (a): Could the authors use symbols that make it visually obvious at first glance which of the DNS and experimental data correspond to each other? E.g. the red filled circle for the experiment together with an empty (red) circle for the corresponding DNS. And maybe a (red) plus for the $Fr = 0$ case as it is separate from all the other data but has the same Ra .

Figure A4 from the response to referee 1 seems worthwhile to be put in the supplement.

The abstract still says "densely populated vortices".

There are still a few typos in the manuscript, here is a non-exhaustive list:

page 1: dynamics plays -> dynamics play

page 2: length H -> height H

$Fr = \Omega^2 d / 2g$ -> $Fr = \Omega^2 d / (2g)$

page 3, right column, first line: consisting of

page 6: bacterial colonies

page 7: RB convection -> RBC (as introduced in the main manuscript)

SI, page 2: long-live -> long-lived

Reviewer #3:

Remarks to the Author:

I remain of my previous opinion that this is outstanding work of the highest international quality, and I strongly support publication. The new (and surprising) dynamics have unambiguously been identified, and the mathematical modelling is appropriate and interesting. It was interesting to read the other referees' reports, and in particular I disagree with ref 1: this work is really exciting and entirely appropriate for publication in Nature C.

Reviewer: 1

RE: Inverse centrifugal effect induced by collective motion of vortices in rotating thermal convection (Resubmission)

In this resubmission (and rebuttal) the authors have provided clear and compelling answers to my points pertaining to the original submission. I want to briefly reply to some answers of the authors to these points.

We thank the Referee for reviewing our manuscript. We are pleased that the Referee is of the view that in our former resubmission (and rebuttal) we have provided clear and compelling answers to his/her previous questions and remarks. Below are our responses (displayed in blue color) to each of the Referee's questions (in black color). All changes made in the revised manuscript are displayed in red.

1. Scope

- I thank the authors for reconsidering the use of "turbulence". The main message of the paper is for a not-quite-turbulent flow state.

- The authors propose compelling examples of natural and technological flows where this study is directly relevant. I agree that my initial assessment was too restrictive. However, the first example of mesocyclones is not readily applicable; these flow structures set up their own centrifugation to act in the force balance (cyclotrophic balance). Centrifugation is here not related to outside effects (i.e. rotation of the Earth) and does not contribute to the displacement /trajectory of mesocyclones.

- On that note, why are some of these examples not briefly mentioned in the paper?

We agree with the Referee that the displacement/trajectory of mesocyclones in the atmosphere is not directly relevant to the centrifugal effects caused by the Earth's rotation, and it is their rapidly rotating structures that set up their own centrifugation-dominant flows in cyclotrophic balance. We remove this example in our discussions of the natural and technological relevance of centrifugation-driven convection.

We have now added these discussions and cited the appropriate references in the second paragraph of the revised manuscript: *"As a primary external force governing the motions of rotating fluids in many natural and industrial flows [13, 14, 15], centrifugal force drives cold, denser fluid radially outward from the rotation axis and warm, lighter fluid inward."*

2. Physical picture of the inner vortex structure & 3. Validation of the results

Thank you for these clarifications and validations! These essential validation steps show that the main message of the paper is not a more basic effect of centrifugation in setting up a meridional circulation inside the cylinder. It is compelling to see that typical vortex drift velocities are significantly larger than the mean flow velocity of the centrifugation-induced meridional circulation.

I am pleased with how the authors have answered my points. This convinced me of the validity of the interpretations of the results. It is important to include the additional validation steps (discussed in the rebuttal) in the supplement. I can now recommend publication. I request the authors to consider my suggestion in item 1.

Following the suggestion from Referee 2, we have included the validation plot on the radial velocities of the meridional circulation and of the vortex motion (Figure 4A in our previous reply) in the revised Supplemental Information.

We take this opportunity to thank the referee once again for the critical and constructive comments and suggestions, which have helped improving both the content and presentation of our manuscript. We are grateful that our manuscript is now recommended for publication in Nature Communications.

Reviewer: 2

The authors have addressed all of my previous concerns.

I was very pleased to see that the authors confirmed their results in cylindrical DNS instead of only Cartesian geometries.

I also appreciated the additional analysis in the upper part of the fluid domain.

These newly added results make the paper even stronger.

I fully recommend the publication of this excellent work in Nature Communications.

We thank the Referee for recommending our work for publication in Nature Communications. We are grateful for all his/her constructive comments and suggestions that have helped us to improve our paper. Below are our responses (displayed in blue color) to each of the comments and questions (in black color). All changes made in the revised manuscript are displayed in red.

The authors may however still want to consider the following minor remarks:

Is the rotation in the experiments in the clockwise direction? This seems to be the case based on figure 1 (e). If so, it should be made explicit somewhere since most studies are anticlockwise by default because Earth rotates anticlockwise.

Also, the DNS are stated to be anticlockwise.

The rotation in the experiment is in the clockwise direction, as we have shown in Figure 1(e). In DNS the rotation direction is anticlockwise. Since the rotation rate is a few orders in magnitude larger than the Earth's rotation, the direction of rotation (clockwise or anticlockwise) in the experiment does not affect the conclusions of the present work.

We have now added descriptions in the Methods section (experimental setup) of the revised manuscript: *"The rotation was set in the clockwise direction with the rotation vector pointing downward (see Fig. 1)."*

A suggestion regarding figure 3 (a): Could the authors use symbols that make it visually obvious at first glance which of the DNS and experimental data correspond to each other? E.g. the red filled circle for the experiment together with an empty (red) circle for the corresponding DNS. And maybe a (red) plus for the $Fr = 0$ case as it is separate from all the other data but has the same Ra .

We thank the Referee for this suggestion. We have made revisions in Figure 3(a) as follows:

- (1) Use red filled circles for the experimental data and empty (red) circle for the corresponding DNS for $Ra=2.0 \times 10^7$.
- (2) Use red plus for the data set with $Fr = 0$ and $Ra=2.0 \times 10^7$.

Figure A4 from the response to referee 1 seems worthwhile to be put in the supplement.

We have added Figure 4A and made discussions in a new section (Supplementary Note 5) in the supplement.

The abstract still says "densely populated vortices".

We have modified this phrase in the abstract as "*densely distributed vortices*"

There are still a few typos in the manuscript, here is a non-exhaustive list:

page 1: dynamics plays -> dynamics play

page 2: length H -> height H

$Fr = \Omega^2 d / 2g$ -> $Fr = \Omega^2 d / (2g)$

page 3, right column, first line: consisting of

page 6: bacterial colonies

page 7: RB convection -> RBC (as introduced in the main manuscript)

SI, page 2: long-live -> long-lived

We thank the Referee for careful reading the manuscript. We have corrected the aforementioned typos in the revised manuscript.